# Cold-induced FOXO1 nuclear transport aids cold survival and tissue storage

Xiaomei Zhang[1,2,3,11], Lihao Ge[4,11], Guanghui Jin[1,2,5,11], Yasong Liu[1,11], Qingfen Yu[6,11], Weizhao Chen[1,2], Liang Chen[1,2], Tao Dong [1,7], Kiyoharu J. Miyagishima [8], Juan Shen[2,9], Jinghong Yang[1,2], Guo Lv[9], Yan Xu[10], Qing Yang[1], Linsen Ye [1,2], Shuhong Yi[1], Hua Li [1], Qi Zhang[2,9,10], Guihua Chen[1,2,9], Wei Liu [2,9] ✉, Yang Yang [1,2,9] ✉, Wei Li [8] ✉ & Jingxing Ou [1,2,9] ✉

Cold-induced injuries severely limit opportunities and outcomes of hypothermic therapies and organ preservation, calling for better understanding of cold adaptation. Here, by surveying cold-altered chromatin accessibility and integrated CUT&Tag/RNA-seq analyses in human stem cells, we reveal forkhead box O1 (FOXO1) as a key transcription factor for autonomous cold adaptation. Accordingly, we find a nonconventional, temperature-sensitive FOXO1 transport mechanism involving the nuclear pore complex protein RANBP2, SUMO-modification of transporter proteins Importin-7 and Exportin-1, and a SUMO-interacting motif on FOXO1. Our conclusions are supported by cold survival experiments with human cell models and zebrafish larvae. Promoting FOXO1 nuclear entry by the Exportin-1 inhibitor KPT-330 enhances cold tolerance in pre-diabetic obese mice, and greatly prolongs the shelf-life of human and mouse pancreatic tissues and islets. Transplantation of mouse islets cold-stored for 14 days reestablishes normoglycemia in diabetic mice. Our findings uncover a regulatory network and potential therapeutic targets to boost spontaneous cold adaptation.

While strategies to make donor cells and tissues into expandable functional organoids provide hope of transplantable tissues[1,2], the improvement of long-term storage techniques remains an essential step toward donor tissue/organ banking. Successful cold storage of live patient tissues for use in disease modeling and drug screening is vital for personalized medicine. Compared to subzero cryopreservation and normothermic continuous perfusion techniques, static cold storage at 0−4 °C is easy to implement and cost-effective[3,4]. However, long-term static 4 °C storage of most human tissues and organs is currently unattainable due to accumulated cellular damage from cold stress and rewarming/ischemia-reperfusion injuries. For example, pancreases and livers can only be preserved at 0−4 °C for about 10 h in University of Wisconsin (UW) solution, the current gold standard for cold-storage of digestive organs[3,5]. Clinically, although blood glucose

[1]Department of Hepatic Surgery and Liver transplantation Center of the Third Affiliated Hospital, Organ Transplantation Institute, Sun Yat-sen University, Guangzhou, China. [2]Guangdong Key Laboratory of Liver Disease Research, the Third Affiliated Hospital of Sun Yat-sen University, Guangzhou, China. [3]Department of Cancer Biology, Dana-Farber Cancer Institute; Department of Cell Biology, Harvard Medical School, Boston, MA, USA. [4]Institute of Psychiatry and Neuroscience, Xinxiang Medical University, Xinxiang, China. [5]State Key Laboratory of Respiratory Disease, National Clinical Research Center for Respiratory Disease, Guangzhou Institute of Respiratory Health, The First Affiliated Hospital of Guangzhou Medical University, Guangzhou, China. [6]Department of Neurology, The Third Affiliated Hospital of Sun Yat-sen University, Guangzhou, China. [7]Department of Surgery, University of Michigan, Ann Arbor, MI, USA. [8]Retinal Neurophysiology Section, National Eye Institute, National Institutes of Health, Bethesda, MD, USA. [9]Guangdong province engineering laboratory for transplantation medicine, Guangzhou, China. [10]Cell-gene Therapy Translational Medicine Research Center, the Third Affiliated Hospital of Sun Yat-sen University, Guangzhou, China. [11]These authors contributed equally: Xiaomei Zhang, Lihao Ge, Guanghui Jin, Yasong Liu, Qingfen Yu. ✉e-mail: lwei6@sysu.edu.cn; yysysu@163.com; liwei2@nei.nih.gov; oujx7@sysu.edu.cn

management tools are currently available for type 1 diabetic patients, human islets replacement therapy (phase 3 trial completed)[6], if available, would be superior owing to their automatic and precise metabolic responses that an insulin administration device lacks[7]. However, this clinical advancement is severely limited by tissue/cell availability[7]. Therefore, prolonging the short shelf-life of pancreases and pancreatic islets can significantly increase the chance to preserve and pool enough islets from multiple donors for transplantation. Although recent advances in tissue vitrification and cryopreservation techniques managed to preserve pancreatic islets for months[8], mechanisms underlying mammalian adaptation to and recovery from freezing or near freezing conditions remain poorly understood, which is vital to improving post-transplantation graft survival and functional restoration.

Studies of hibernators, such as 13-lined ground squirrels (TLGSs), who are naturally cold adaptive, can provide unique insights for cold protection and inspire novel translational strategies[9–13]. For example, in hibernating TLGS the small ubiquitin-like modifier (SUMO) machinery was found to enhance protein-SUMO covalent conjugation, which inspired later studies suggesting that protein SUMOylation is a conserved response in human cell lines to reprogram transcription and enhance tolerance to fluctuation in temperature or oxygen and nutrient supplies[14–16]. Additionally, many proteins also contain SUMO-interacting motifs (SIM)[17,18] that partner with SUMOylated proteins via non-covalent binding. How these SIM-containing proteins work with the SUMO machinery and what roles they play in cellular adaptation to cold stress remain little explored.

As for non-hibernating species, studies show that newborns of various mouse strains can recover from hours of ultra-profound hypothermia (body temperature at 0–8 °C), suggesting that non-hibernator, at early developmental stage, may also possess cold-resistance feature that might be lost as animals mature[19,20]. Consistently, good tolerance to cold exposure has been shown in human embryonic stem cells (hESCs)[21]. These examples prompted us to seek common molecular mechanisms underlying cold adaption in various species. Such knowledge may guide the design of better storage strategies to achieve longer-term static cold preservation of donor organs/tissues, and together with the cryopreservation and machine perfusion techniques lay the foundation for large-scale banking of live donor cells, tissues and organs.

## Results

### FOXO1 supports cell survival from prolonged cold exposure

ATAC-seq was first used to probe genome-wide chromatin accessibility change in H1 ESCs during 4-h exposure at 4 °C (Fig.1a, b and Supplementary Fig. 1a, b). We found that FOXO1-binding motif is among the top enriched transcription factor-binding motifs at 4 °C. FOXO family transcription factors are known to respond to many biological stressors[22]. We thus probed whether FOXO1 could be a key cellular responder to cold stress with enhanced chromatin binding upon cold exposure. Indeed, cell fractionation and western blotting demonstrated a cold-induced FOXO1 translocation from cytosol to nuclei in human ESCs (Fig.1c). This was confirmed by confocal and stimulated emission depletion (STED) imaging of FOXO1 immunofluorescent signals (Fig.1d, e). A FOXO1 inhibitor (FOXO1-i) that prevents FOXO1 from DNA binding[23], exacerbated 48 h 4 °C exposure-induced cell death in H1 ESCs and induced pluripotent stem cell (iPSC)-derived neurons from a human infant donor, and led to poor cell physiology marked by deteriorated colony morphology and pluripotency in H1 ESC cultures passaged following shorter (12-h) 4 °C exposure (Fig.1e and Supplementary Fig. 1c–e), suggesting that such enhanced chromatin-binding of FOXO1 is required for cold survival and recovery. Such temperature-induced FOXO1 relocation feature was also observed in TLGS (Fig.1f, g), indicating that it is a conserved temperature-sensing mechanism.

### FOXO1 regulates the transcription of novel target genes during rewarming

We then explored how FOXO1, when enters nuclei, shapes the transcriptional landscape of H1 ESCs as they face drastic temperature changes. Consistent with the ATAC-seq analysis, FOXO1 CUT&Tag[24] assays reveal a number of genes with enhanced binding by FOXO1 at 4 °C (Fig. 2a, b, Supplementary Fig. 2a–c and Supplementary Data 1). Most of these genes have not been previously recognized as FOXO1-regulated targets. Further, we used RNA-seq to analyse the transcriptional changes of these genes. Interestingly, the H1 ESC transcriptome barely changed following 4-h at 4 °C. In contrast, during 2-h recovery at 37 °C, there are 2742 differentially expressed genes (DEGs), among which ~10% (247) were also identified by FOXO1 CUT&Tag assays, accounting for ~20% genes with enriched FOXO1-binding at 4 °C (Fig. 2c and Supplementary Data 1). These analyses demonstrate the significance of FOXO1 and other regulators (Fig. 1b) in reprogramming the H1 transcriptome during recovery from cold stress. To understand how FOXO1 mediates H1 cell physiology during temperature adaptation, enriched pathways from these 247 FOXO1 target genes (Fig. 2d) were compared to that impacted by cold exposure and rewarming the most in H1 cells (Supplementary Fig. 2d, e and Supplementary Data 2). Accordingly, FOXO1 contributes to transcriptomic changes that affect rhythmic processes, protein phosphorylation and RHO GTPase activities following rewarming.

### RANBP2/IPO7/XPO1 mediates temperature-dependent FOXO1 translocation

Next, we probed the mechanism of temperature-dependent FOXO1 translocation. Dephosphorylation and Sirtuin-mediated deacetylation are the known pathways that can retain FOXO proteins that are already in cell nucleus[25,26]. However, in multiple cell systems that we examined, these mechanisms do not support cold-induced translocation of FOXO1 from cytosol to nucleus. For example, in H1 ESCs we found initial increase in FOXO1 T24, S256 and S329 phosphorylation at 4 °C (Supplementary Fig. 3a), which should favor FOXO1 nucleus-exiting. However, endogenous phosphorylated FOXO1 proteins or over-expressed mutants mimicking a constitutively phosphorylated form of FOXO1 were accumulated in the nuclei upon 4 °C exposure (Supplementary Fig. 3b, c), suggesting that cold stimuli override the nucleus-exiting effect of FOXO1 phosphorylation. Additionally, siRNA-interfering the expression of Sirtuins failed to alter the temperature-dependent FOXO1 dynamics in H1 ESCs, and K294 acetyl-FOXO1 proteins were also accumulated in H1 nuclei at 4 °C (Supplementary Fig. 3d–g). These results indicated unidentified mechanisms governing cold-induced FOXO1 nuclear entry and function. Here, we discovered that this process is promoted by the SUMO1 E3 ligase subunit RANBP2 and nuclear importer Importin-7 (IPO7), whilst antagonized by nuclear exporter Exportin-1 (XPO1).

RANBP2 is also a nuclear pore protein that mediates nuclear transport via the Importin/Exportin system[27,28]. FOXO1, IPO7 and XPO1 contain putative[29,30], conserved SIM or SUMOylation sites (Supplementary Fig. 4a). A modest increase of SUMOylated proteins was found in H1 ESCs 1-h after 4 °C incubation (Supplementary Fig. 4b). Thus, we hypothesized that the SUMO machinery may mediate the nuclear transporters and hence affect FOXO1 transport when temperature changes. Accordingly, a significant increase of RANBP2 and IPO7, but not XPO1 or other main SUMO ligases[31,32] was identified following cold exposure (Supplementary Fig. 4c, d). Moreover, siRNA knock down (KD) of *RANBP2* and *IPO7* expression blocked cold-induced FOXO1 nuclear entry, while KD of *XPO1* resulted in FOXO1 nuclear retention regardless of the temperature (Fig. 3a and Supplementary Fig. 4e). In contrast, KD of other E3 ligase subunits, such as RANGAP1 or UBC9[28], produced little effect on FOXO1 subcellular distribution (Supplementary Fig. 4f).

Furthermore, proximity ligation assay (PLA) revealed a stronger FOXO1/IPO7 interactions at 4 °C, in favor of nuclear importing; at 37 °C

however, such interaction weakened, whilst FOXO1/XPO1 interaction became stronger, tipping the balance towards nuclear exporting (Fig. 3b, c). Immunoprecipitation (IP) and western blotting confirmed the temperature-dependent interaction patterns of FOXO1 with either XPO1 or IPO7 (Fig. 3d). Accordingly, at 4 °C, IPO7/RANBP2 and IPO7/

SUMO1 interactions were stronger, while at 37 °C, XPO1/RANBP2 and XPO1/SUMO1 interactions were stronger (Fig. 3e, f and Supplementary Fig. 4g). In contrast, SUMO2/3 interactions with FOXO1, IPO7 and XPO1 were not temperature-dependent (Supplementary Fig. 4h). Next, to assess the effect of IPO7 SUMOylation in a human cell model easier to be

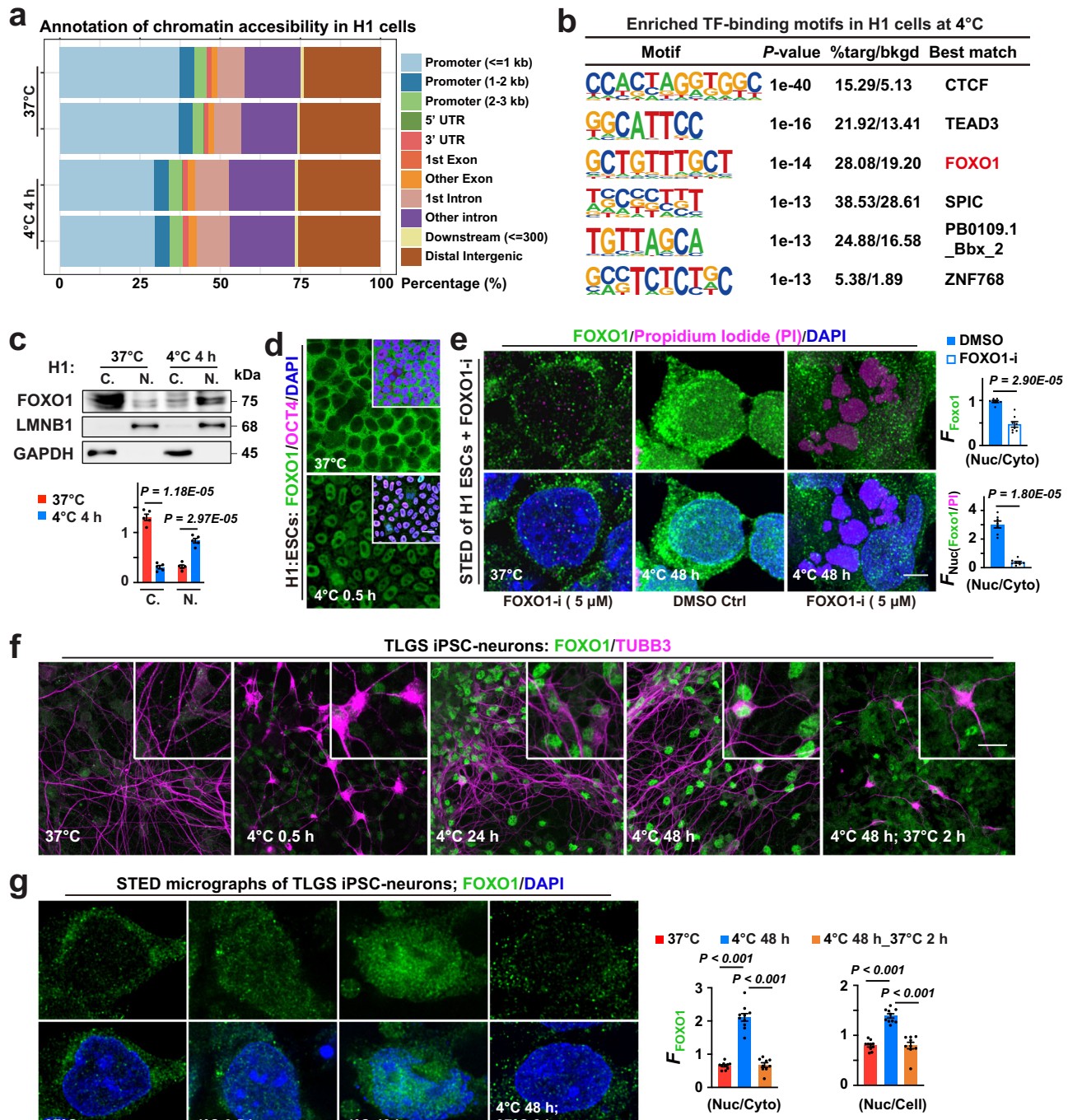

**Fig. 1 | FOXO1 and cell cold adaptation. a** Stacked bar plots showing genomic annotation of chromatin accessible regions in different groups. **b** Binding motifs of transcription factors (TF) enriched in H1 cells following 4-h incubation at 4 °C. **c** Up: Immunoblots of FOXO1 proteins in cytosolic (C.) and nuclear (N.) fractions, conditions annotated; down: FOXO1 protein levels (n = 5 experiments). **d** Confocal images of FOXO1, pluripotency marker OCT4 and nuclear DNA (DAPI) staining in human embryonic stem cells (H1 ESCs) at indicated conditions (n = 10 experiments). **e** Left: Stimulated emission depletion (STED) micrographs detailing FOXO1, Propidium Iodide (PI; to stain dead cells) and DAPI signals in H1 cells at indicated conditions; right: Nuc $F_{FOXO1}$ versus Cyto $F_{FOXO1}$, or versus $F_{PI}$ (n = 7 and 8

experiments for DMSO and FOXO1-i, respectively); FOXO1-i, FOXO1 inhibitor. **f** Confocal images of FOXO1 and neuronal marker TUBB3 in TLGS iPSC-derived neurons at indicated conditions. **g** Left: STED images showing FOXO1 proteins and nuclear DNA (DAPI-stained) in TLGS iPSC-neurons at indicated conditions; right: the mean intensity of FOXO1 fluorescence ($F_{FOXO1}$) in the nucleus (Nuc) versus that in the whole cell (Cell) or the cytosol (Cyto) was analyzed (from left to right, n = 10, 10, 9 images from 5 experiments). Data are shown as mean and SEM. Statistics: two-tailed Student's t test (**c, e**) and one-way ANOVA followed by Tukey's test (**g**). Scale bars: 20 μm (**d, f**) and 2.5 μm (**e, g**). Source data are provided as a Source Data file.

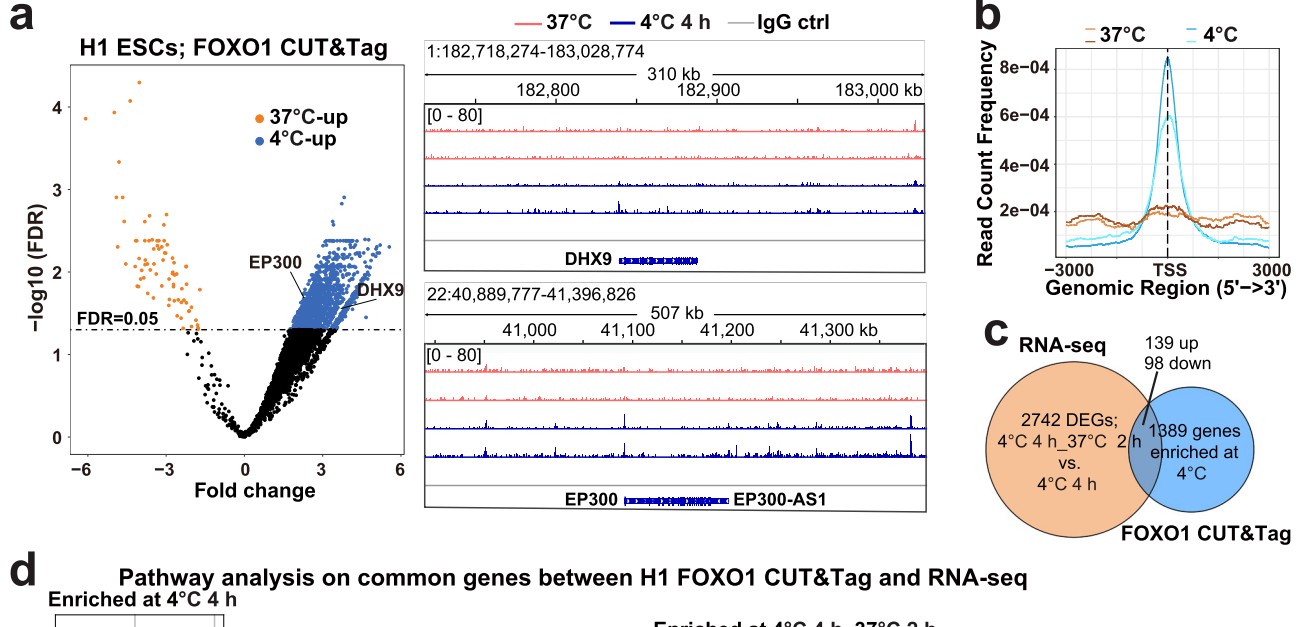

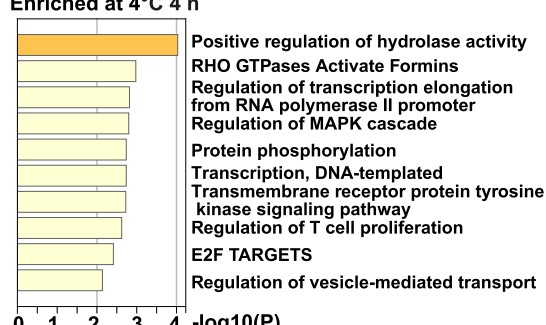

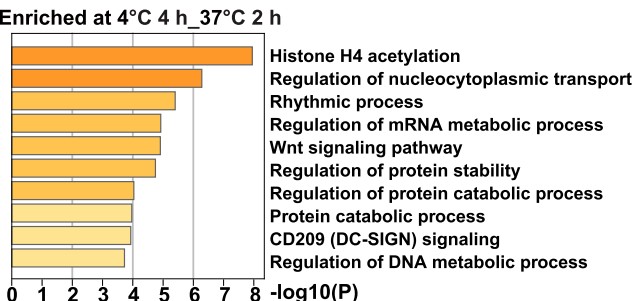

**Fig. 2 | Temperature-induced differential FOXO1-DNA binding and transcriptomic changes in H1 ESCs. a** Left: fold change vs. -log$_{10}$ FDR-corrected Q-value for differential FOXO1-binding at 37 °C or 4 °C 4 h in H1 ESCs revealed by CUT&Tag-seq; right: IGV (Integrative Genomics Viewer) snapshots showing FOXO1-binding signals around the DHX9 and EP300 loci (right); kb, kilobase pairs. **b** Line plots summarizing the mean of FOXO1 CUT&Tag signals within 3 kb up- and down-stream of gene transcription start sites (TSS) in H1 ESCs at 37 °C and 4 °C 4 h; kb, kilobase pairs. **c** Venn diagram indicating the overlap between differentially expressed genes (DEGs) upon rewarming and 4 °C-induced FOXO1 differentially bound genes in H1 ESCs. **d** Enrichment analysis on the overlapped genes from (**c**); left: Enrichment analysis on genes that had higher levels of transcripts at 4 °C 4 h compared to 4 °C 4 h_37 °C 2 h (rewarmed); right: Enrichment analysis on genes that had higher levels of transcripts at the rewarmed stage compared to 4 °C 4 h. *P* values in (**d**) were calculated in Metascape.

transfected than H1 ESCs, mutant IPO7 proteins with K517R disruption of the predicted SUMOylation site were overexpressed in human ARPE-19 cells. PLA showed that this site mutation abrogated the importer's binding with FOXO1 at 4 °C (Fig. 3g). These results support a previously unappreciated mechanism that RANBP2/IPO7/XPO1 form a SUMOylation-dependent, temperature-sensing transport system to govern FOXO1 cellular localization and subsequent transcriptional activities.

### XPO1 SUMOylation and FOXO1 SIM promotes FOXO1 cytoplasmic localization at 37 °C

We further investigated the role of SUMO1-SIM interactions in the process of IPO7/XPO1-mediated FOXO1 translocation. Firstly, we aimed to mutate the putative FOXO1 SIM (Supplementary Fig. 4a) to probe its role in FOXO1 translocation. As FOXO1 is essential to hESC maintenance[33] and hence FOXO1 mutation attempts were not suitable for H1, we instead performed CRISPR/Cas9 editing[34] in ARPE-19 cells to disrupt the endogenous FOXO1 SIM domain (Supplementary Fig. 5a). In these mutant cells, FOXO1/SUMO1 interaction became temperature-insensitive (Supplementary Fig. 5b) and FOXO1-minus SIM proteins predominantly localized to the cell nucleus regardless of temperature (Fig. 4a), revealing that FOXO1 SIM is required for FOXO1 nuclear export. IP and western blotting verified that disrupting FOXO1 SIM significantly strengthened FOXO/IPO7 but weakened FOXO1/XPO1

interaction (Fig. 4b). Importantly, nuclear localization of FOXO1-minus SIM proteins still requires RANBP2 and IPO7 (Supplementary Fig. 5c). Consistently, overexpressing *FOXO1-minus SIM* in H1 ESCs also rendered nuclear localization of these mutant proteins independent of temperature (Supplementary Fig. 5d). Either FOXO1-minus SIM mutation or XPO1 mutations that eliminate its putative SUMOylation sites (K752R and K957R) prevented FOXO1/XPO1 interaction at 37 °C (Fig. 4c), proving that XPO1 SUMOylation and FOXO1 SIM are crucial to their protein-protein interactions.

FOXO1 contains the classical nuclear localization sequence and nuclear export signal (NLS and NES)[25,35] (Supplementary Fig. 5a). It is unclear if these domains affect temperature-dependent FOXO1 transport. Apparently (Fig. 4d), the overexpressed, NLS/NES-truncated FOXO1 (in-frame deletion of amino acids 245-274, and 371-380) mutant proteins remained to be predominantly in the cytoplasm of ARPE-19 cells during 4 °C incubation. Meanwhile, overexpressed FOXO1 and FOXO1-minus NLS/NES proteins had similar interaction patterns with XPO1 or IPO7 at 37 °C and 4 °C (Fig. 4e). Thus, FOXO1 NLS/NES do not overtly contribute to the temperature-sensitive FOXO1-XPO1/IPO7 interactions. However, FOXO1 NLS is essential to its nuclear import.

Consistently, FOXO3 proteins adopt constitutive nuclear localization in ARPE-19 cells, whilst adding FOXO1 SIM to FOXO3 rendered it cytosolic distribution at 37 °C and nuclear accumulation at 4 °C (Fig. 5b

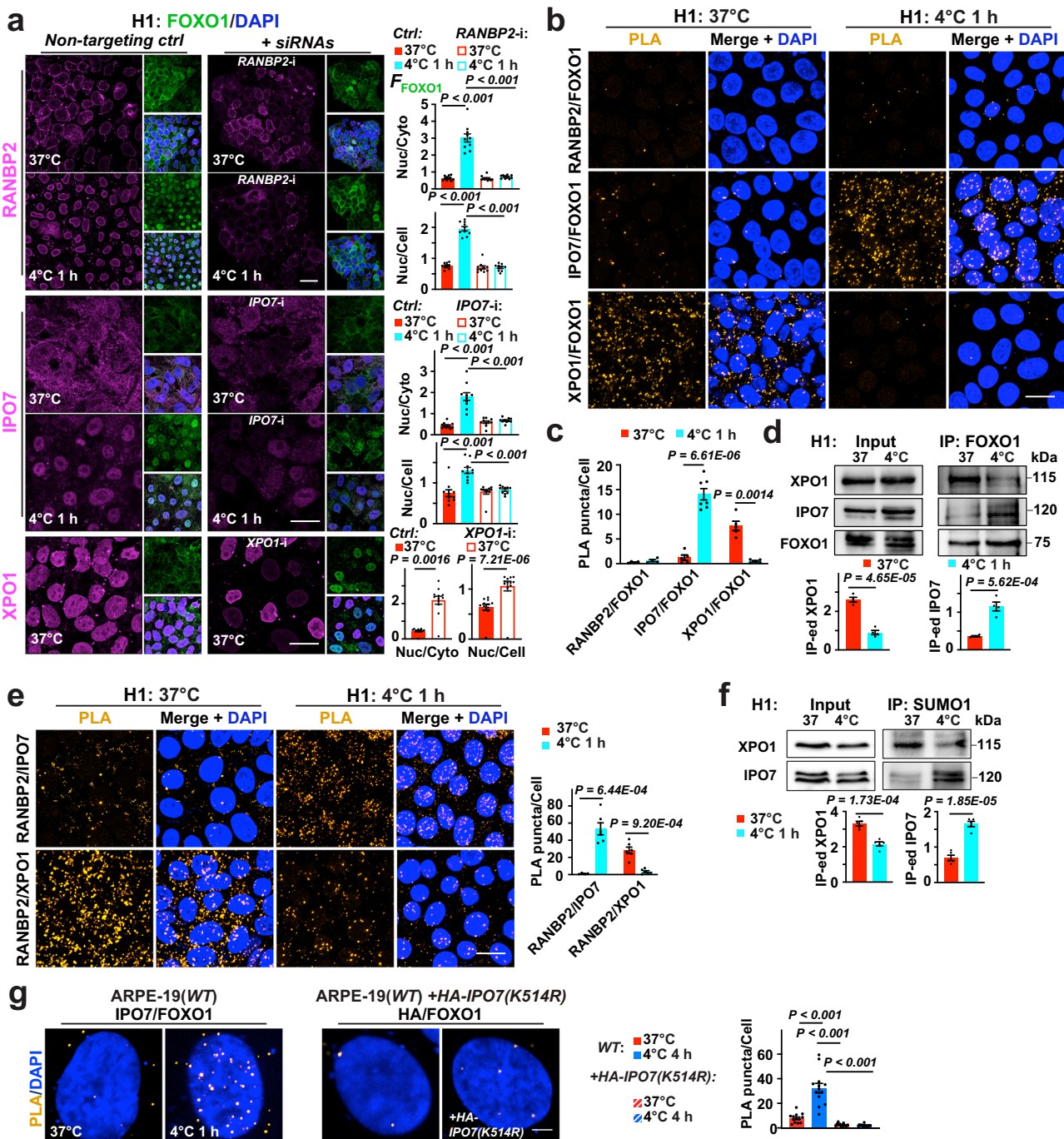

**Fig. 3 | Temperature-mediated FOXO1 transport dependent on RANBP2, XPO1 and IPO7 SUMOylation. a** Left: Confocal images of H1 ESCs with annotated treatments; FOXO1, DAPI were co-stained with RANBP2, Importin-7 (IPO7) or Exportin-1 (XPO1); right: Nuclear $F_{FOXO1}$ versus cytosolic or whole-cell $F_{FOXO1}$ ($n = 10$, 10 and 12 images from 5 experiments for *RANBP2*, *IPO7* and *XPO1* RNA interference, respectively). **b** Confocal images of proximity ligation assay (PLA) on proteins of interest in H1 ESCs at indicated conditions. **c** Average counts of PLA puncta per cell from (**b**) (from left to right, $n = 4$, 4, 5, 7, 5 and 5 images from 3 experiments). **d** Up: immunoblots of XPO1, IPO7 and FOXO1 in total protein extracts (Input) and FOXO1 immunoprecipitated (IP-ed) fractions from H1 cells at annotated conditions; down: signal intensities normalized ($n = 4$ experiments). **e** Left: confocal images of PLA on

proteins of interest in H1 ESCs at indicated conditions; right: average counts of PLA puncta per cell (from left to right, $n = 4$, 6, 6 and 6 experiments). **f** Up: immunoblots of XPO1 and IPO7 in Input and SUMO1 IP-ed fractions from H1 cells at annotated conditions; down: signal intensities normalized ($n = 5$ experiments). **g** Left: confocal images of PLA on FOXO1 interaction with endogenous IPO7 ($n = 13$ images from 3 experiments), or with overexpressed, HA-tagged IPO7 with a K-R mutation on its SUMOylation site in ARPE-19(*WT*) cells ($n = 12$ images from 3 experiments); *WT*, *wild-type*; right: average counts of PLA puncta per cell. Data are shown as mean and SEM. Statistics: one-way ANOVA followed by Tukey's test (**a, g**) and two-tailed Student's *t* test (**c–f**). Scale bars: 20 μm (**a, b, e**) and 2.5 μm (**g**). Source data are provided as a Source Data file.

−d). Taken together, our findings reveal essential characteristics of a cell autonomous cold-adaptive interactome (Fig. 5e): a SUMO ligase/ nuclear pore component – RANBP2, its transporter partners IPO7 and XPO1 with SUMOylation sites, and a cargo FOXO1 with SIM.

**Targeting the XPO1-FOXO1 axis improves cold survival in vivo**
Next, we investigated whether this FOXO1-mediated cold adaptive response occurs in vivo. We first tested cold survival in zebrafish larvae, as this ectothermic model organism is cold intolerant. Indeed,

zebrafish also showed cold-induced Foxo1 protein nuclear entry in early development, and when *foxo1a* expression was repressed, cold-exposed larvae more frequently manifested pericardiac edema and

severe and irreversible abnormal body curvatures (Supplementary Fig. 6a–d and Supplementary Movie 1). Also, FOXO1 proteins were enriched in TLGS cell nuclei of multiple tissues and organs during

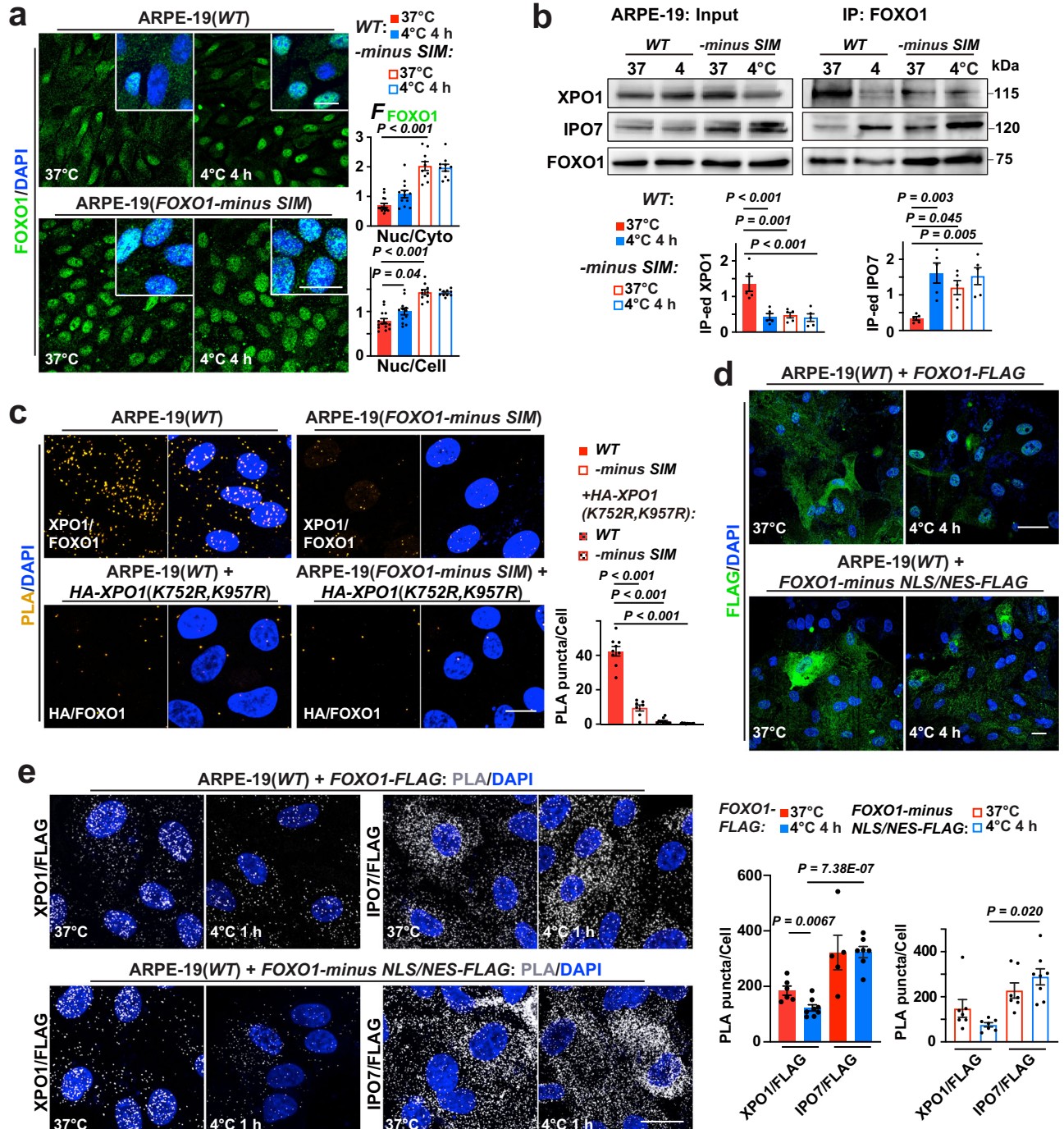

**Fig. 4 | FOXO1 nuclear export determined by FOXO1 SUMO-interacting motif (SIM) and XPO1 SUMOylation. a** Left: confocal images of FOXO1 and DAPI in ARPE-19(*WT*) and ARPE-19(*FOXO1-minus SIM*) at annotated conditions; right: $F_{FOXO1}$ ratio analyzed (from left to right, $n$ = 14, 12, 10 and 10 images from 5 experiments). **b** Up: Input and FOXO1 IP-ed fractions from ARPE-19(*WT*) and ARPE-19(*FOXO1-minus SIM*) cells at annotated conditions were immunoblotted against XPO1, IPO7 and FOXO1; down: signal intensities normalized ($n$ = 5 experiments). **c** Left: confocal images of PLA on FOXO1 interaction with endogenous XPO1, or with overexpressed, HA-tagged XPO1 with both SUMOylation sites replaced with an arginine residue (K-R mutation) in ARPE-19(*WT*) and ARPE-19(*FOXO1-minus SIM*) cells at 37 °C (from left to right, $n$ = 9, 7, 11 and 11 images from 3 experiments); right: average counts of PLA

puncta per cell. **d** Confocal images of FLAG tag and DAPI staining in ARPE-19(*WT*) cells overexpressing FLAG-tagged, FOXO1 or mutant FOXO1 deleted of both NLS and NES domains (FOXO1-minus NLS/NES; in-frame deletion of amino acids 245-274 and 371-380) at annotated conditions ($n$ = 3 experiments); NLS, nuclear localization sequence; NES, nuclear export signal. **e** Left: confocal images of PLA on indicated interacting pairs in ARPE-19(*WT*) cells overexpressing FLAG-tagged, FOXO1 or FOXO1-minus NLS/NES at annotated conditions; right: average counts of PLA puncta per cell (from left to right, $n$ = 6, 8, 5, 7, 7, 7, 7 and 8 images from 3 experiments). Data are shown as mean and SEM. Statistics: one-way ANOVA followed by Tukey's test (**a**–**c**) and two-tailed Student's $t$ test (**e**). Scale bars: 20 μm. Source data are provided as a Source Data file.

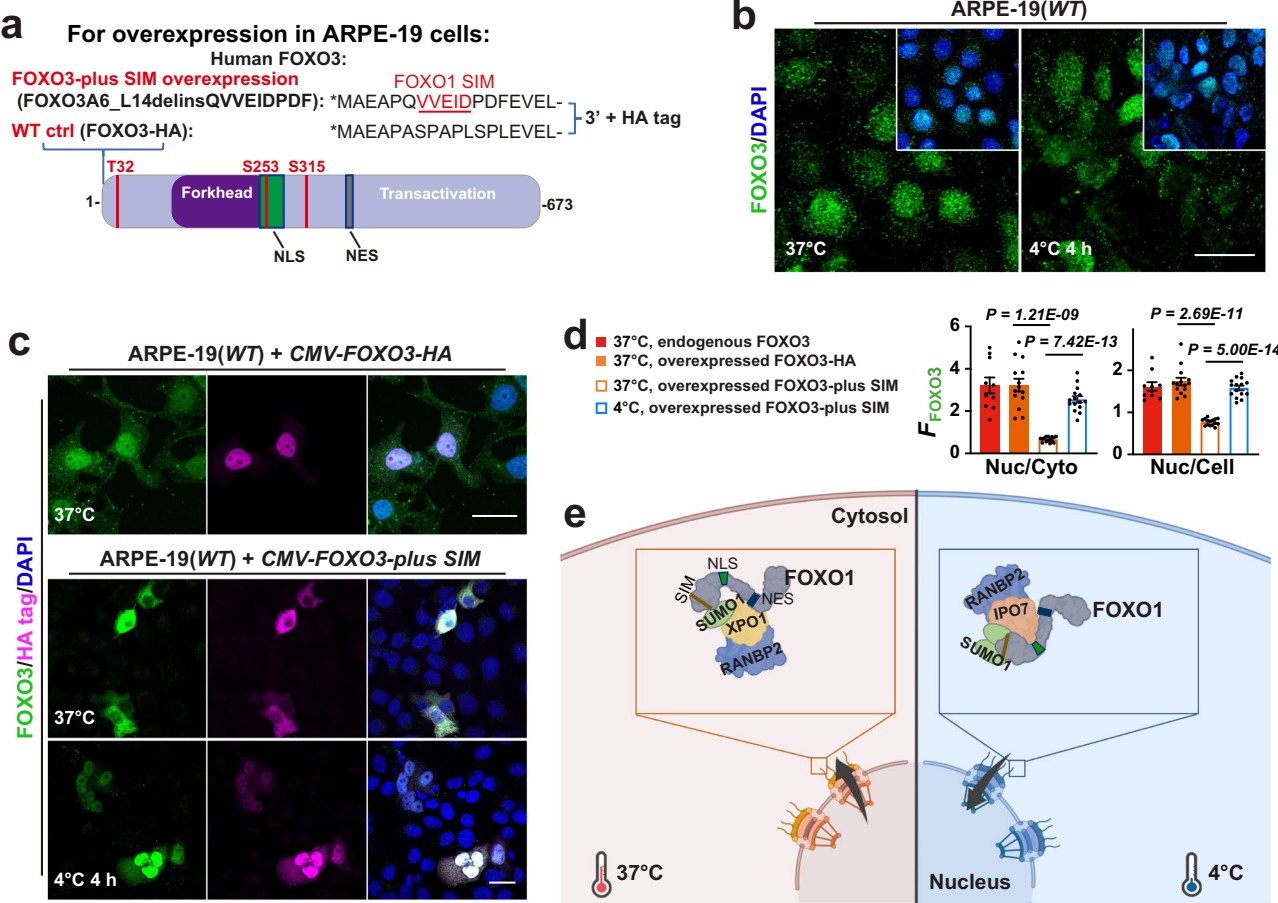

**Fig. 5 | FOXO1 SIM as a Nuclear Exit Signal for FOXO3. a** Schema of FOXO3 mutant construct that contains the FOXO1 SIM. **b** Confocal images of FOXO3 and DAPI staining in ARPE-19(*WT*) cells at indicated conditions (*n* = 6 experiments). **c** Confocal images of FOXO3, HA tag and DAPI staining in ARPE-19 cells over-expressing *FOXO3-plus SIM-HA* at indicated conditions. **d** $F_{FOXO3}$ ratio (from left to right, *n* = 10, 14, 15 and 15 images from 5 experiments). **e** A model for temperature and FOXO1 SIM-mediated FOXO1 transportation. The illustration was based on a template created with BioRender.com. Data are shown as mean and SEM. Statistics: two-tailed Student's *t* test (**d**). Scale bars: 20 μm (**b**, **c**). Source data are provided as a Source Data file.

hibernation or following storage at 4 °C, while FOXO3 did not show such dynamics, suggesting a specific role of FOXO1 in cold adaptation during hibernation (Supplementary Fig. 6e–g).

Cold-exposure treatment has demonstrated its efficacy in ameliorating obesity and diabetes-related symptoms in humans and mice[36–40]. However, acute cold exposure also imposes tremendous stress on prediabetic obese mice, which was used in this study (Supplementary Fig. 7a) to test the effect of manipulating the FOXO1 localization in cold tolerance. We applied an FDA-approved XPO1-inhibitory drug, KPT-330[41], to promote FOXO1 nuclear accumulation (Fig. 6a, b). One injection of 3 mg/kg KPT-330 aided obese middle-aged mice to adapt and survive for 10 days of cold exposure at 4 °C, while vehicle-control group suffered 75% fatality within the first 3 days of cold exposure (Fig. 6c). When cold exposure started, KPT-330 treatment appeared to facilitate heat production, $O_2$ consumption and $CO_2$ production; quantitative PCR assays on the expression of thermogenesis-related genes[42,43] revealed only mild further upregulation of *Cidea* in epididymal white adipose tissues (eWAT) and skeletal muscles, and *Ppargc1a* in skeletal muscles; more weight loss in the first 2 days of cold exposure, more total fat loss and slight increase in lean weight were observed in the KPT-330 treatment group 2 days after recovery, while no significant difference in food-intake or activity between the treatment and control groups was observed throughout the experiments (Fig. 6d–f and Supplementary Fig. 7b–f). Similar effects of KPT-330 treatment were seen in healthy young adult mice (Supplementary Fig. 7g).

To explore KPT-330 treatment effects other than thermogenesis, samples of eWAT, brown adipose tissues (BAT), skeletal muscles and liver from these obese mice with KPT-330 or DMSO vehicle injection were collected 6 h after 4°C exposure and subjected to transcriptomic analysis. DEGs common to these 4 tissues/organs may represent universal effector genes of KPT-330 treatment (Fig. 6g and Supplementary Data 3). Enrichment analysis revealed that other than cellular responses to stress or starvation, the most notable impact of KPT-330 treatment on cold tolerance is in modulating immune responses and cytokine signaling (Fig. 6h and Supplementary Fig. 7h-k).

### Targeting the XPO1-FOXO1 axis aids pancreatic islets cold storage and transplantation

Encouraged by these results, we aimed to develop a better cold storage solution for donor organs and tissues such as pancreas and pancreatic islets, which have the shortest shelf life among transplantable organs/tissues. We found that pre-treatment of KPT-330 was sufficient to promote FOXO1 nuclear entry in pancreatic tissues cold-stored for 48 h (Supplementary Fig. 8a). This KPT-330 pretreatment, in combination with cold storage in hibernation solution (HS, see Methods for details) that well-preserved SUMO proteins and nuclear transporters (Supplementary Fig. 8b), drastically reduced cold-triggered cell death (Fig. 7a). In contrast, inhibiting the SUMO machinery with 2-D08 or Ginkgolic Acid prevented FOXO1 nuclear entry and led to poor storage outcomes (Fig. 7b and Supplementary Fig. 8c, d). Importantly, this

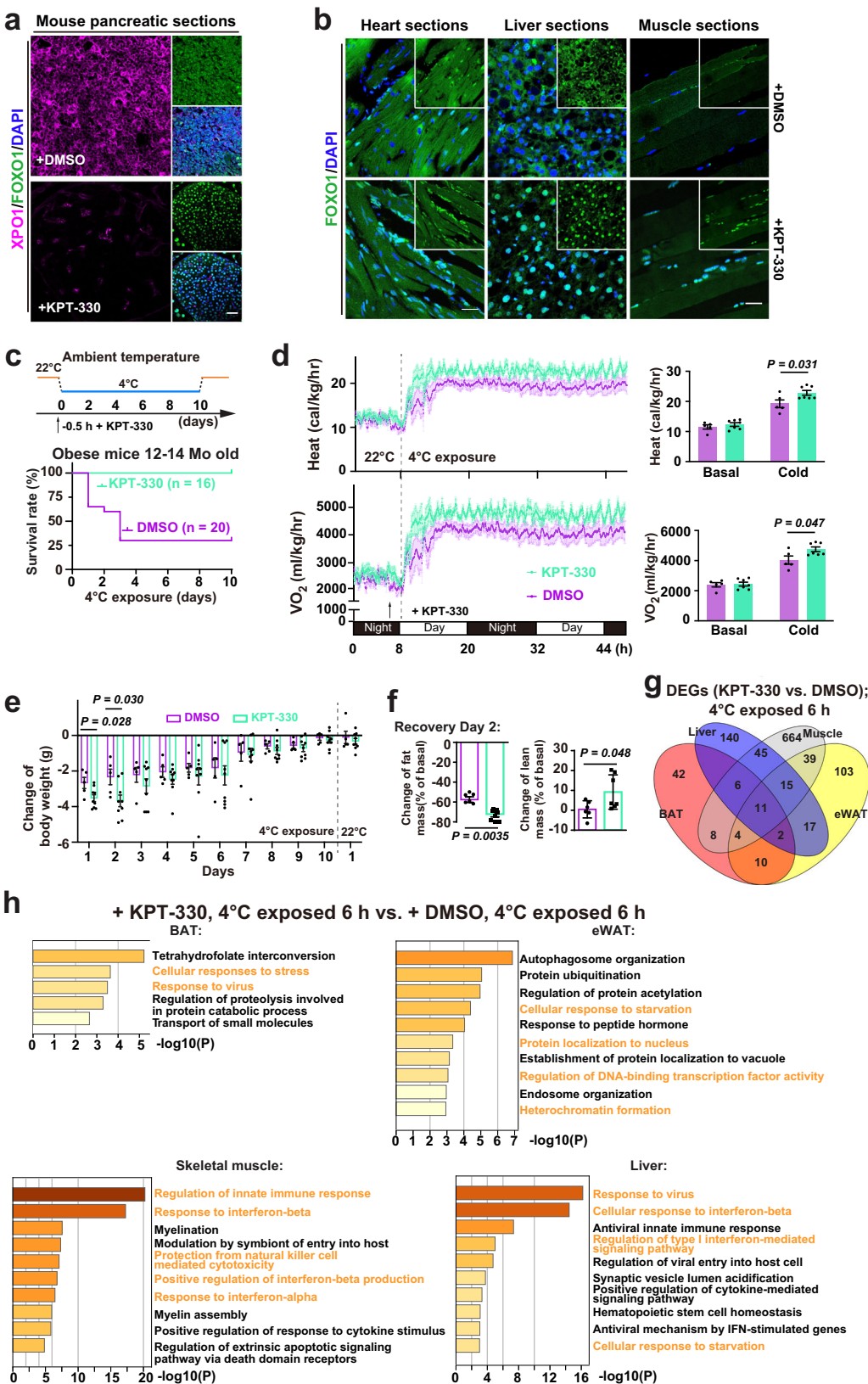

superior cold storage strategy failed to preserve islets from mice with β cell-specific KD of FOXO1 (*Ins1-Foxo1-KD*) (Fig. 7b and Supplementary Fig. 8d), confirming that FOXO1 functions are vital to cold protection. Measured by normalized oxygen consumption rate (OCR)[44,45] and glucose-stimulated insulin secretion (GSIS), mouse islets cold-stored in UW solution for 5–7 days showed poor preservation and GSIS

following 24-h in vitro recovery; protease inhibitors, which have been shown to provide cold-protection in our previous work[9], were also added to HS (HS + KPT-330 and protease inhibitors; K&P) to cold-store islets for up to 14 days; then, islets stored in HS + K&P demonstrated GSIS within 2-h in vitro rewarming, whilst islets stored in UW solution suffered from severe cell death and tissue disintegration (Fig. 7c–f, and

**Fig. 6 | Activating FOXO1 nuclear entry to enhance cold survival in obese prediabetic mice. a** Confocal images of XPO1, FOXO1 and DAPI staining in obese mice (12-14 mo) pancreatic sections from indicated conditions ($n = 5$ mice from 5 experiments); KPT-330, XPO1 inhibitor. **b** Confocal images of FOXO1 and DAPI in obese mice heart, liver and skeletal muscle sections from indicated conditions ($n = 5$ mice from 5 experiments). **c** Survival rate of obese mice at indicated conditions; KPT-330, XPO1 inhibitor; DMSO is the solvent control. **d** Left: from a representative mouse energy expenditure experiment, heat and oxygen consumption ($VO_2$) of obese mice with an intraperitoneal injection of KPT-330 (3 mg/kg) ($n = 7$) or DMSO ($n = 5$) prior to 4 °C exposure; right: average heat and $VO_2$ under basal and cold-exposed conditions. **e** Daily changes of mouse body weight during 4 °C exposure (KPT-330, $n = 9$; DMSO, $n = 5$). **f** Mouse body mass composition following 10-day cold exposure at indicated conditions (KPT-330, $n = 8$; DMSO, $n = 6$). **g** Venn diagram showing the numbers of overlapped DEGs among indicated tissues from obese mice after 6 h of 4 °C exposure; also see Supplementary Data 3. **h** Enrichment analysis on DEGs upregulated in brown adipose tissues (BAT), epididymal white adipose tissues (eWAT), skeletal muscles and liver at indicated conditions. Data are shown as mean and SEM. Statistics: two-tailed Student's $t$ test (**d–f**); $P$ values in (**h**) were calculated in Metascape. Scale bars: 20 μm (**a**, **b**). Source data are provided as a Source Data file.

Supplementary Fig. 8e, f). Mouse islets stored and protected by this new formula were transplanted into streptozotocin-induced diabetic mice and were able to lower the blood glucose levels of the recipient mice, demonstrating robust in vivo functional recovery of these islets (Fig. 7e, f). Human donor pancreatic tissues (up to 2 days) and islets (up to 14 days) isolated from them had normal morphology and displayed functional GSIS (Fig. 8a–c and Supplementary Fig. 9). While further work is required to establish whether the protocol can produce successful storage of human tissues for actual transplantation, our findings provide a practical solution that may facilitate future islet banking and transplantation in diabetic patients.

## Discussion

We revealed, from ectothermic vertebrate to mammalian hibernator to human organ, and from cell cold response to mouse cold survival to tissue cold storage, that FOXO1 is a key adaptive factor recognized by the nuclear transport system via SUMO-interactions in a temperature-dependent fashion. We demonstrate that SIM is a previously unappreciated motif recognized by SUMOylated XPO1 promoting a cytoplasmic localization of FOXO1. SIM may serve as a cargo code of subcellular redistribution for transporter proteins during the adaptation to and recovery from drastic temperature changes. It will be interesting to evaluate to what extent other known post-translational modifications of FOXO1 may impact the conformational changes of FOXO1 and hence the interactions between FOXO1 and transporter proteins during temperature changes. Other transcription factors containing SIM can be similarly investigated for their roles in cell adaptation, metabolic reprogramming and stress responses. In addition, SIM, nuclear transporters, and their SUMOylation sites may be attractive pharmaceutical targets that warrant further explorations.

Although the importance of FOXO transcription factors have been well recognized in metabolic regulations, development and aging, immunity and stress response[22,46–53], the complex roles of FOXO1 in cold tolerance through modulating glucose utilization and thermogenesis are just emerging[54,55]. Notably, transcriptomic analysis on zebrafish and tilapia inferred a pivotal role of *foxo3b* in cold tolerance[56], and high temperature upregulates *foxo1* and *foxo3* in the skeletal muscle of red cusk-eel[57], hinting the involvement of Foxo3 functions in temperature adaptation in ectothermic species. Note that our H1 ESC FOXO1 CUT&Tag assays revealed numerous genes bound by FOXO1 at 4 °C but not at 37 °C (Fig. 2 and Supplementary Data 1). Most of these genes have not been recognized as FOXO1 target genes, hence conventional RNA-seq and pathway enrichment analysis may be unable to identify real master transcriptional pathways in mediating adaptation to temperature changes. In addition, technical difficulty in isolating cells from whole tissues to perform CUT&Tag and ATAC-seq assays remains, making it challenging to address the tissue-specificity issue. In this regard, stem cell-derived cell types of interest and in vitro temperature-challenge models may provide a feasible and reliable platform to evaluate the functions of FOXO3 and other transcription factors/nuclear proteins such as CTCF and TEAD3 (Fig. 1b) during temperature adaptation and recovery.

To demonstrate the importance of the XPO1/FOXO1 axis in cold adaptation and survival, here the FDA-approved selective XPO1 inhibitor KPT-330 was used. Originally designated as an anti-cancer drug[58], KPT-330 showed its potent efficacy in cold survival and enhancing cold-induced energy expenditure and fat loss in obese prediabetic mice, and in prolonging cold storage of human and mouse pancreases and isolated islets. Although KPT-330 treatment apparently led to similarly elevated heat production, $O_2$ consumption and $CO_2$ production in both healthy young adult mice and obese mice, their cold-stimulated stress and thermogenic responses may not be the same and await in-depth comparative studies. Intriguingly, the most notable impact of KPT-330 treatment on cold tolerance of the obese mice appeared to be in modulating immune responses (Fig. 6h and Supplementary Data 3). How the immune system affects cold adaptation and heat production (Fig. 6d and Supplementary Fig. 7g) in normal and health-compromised conditions is worth addressing in the future. Interestingly, KPT-330 treatment enhanced the expression of thermogenic genes *Ppargc1a* and *Cidea* in the skeletal muscle samples of obese mice following 4 °C exposure (Supplementary Fig. 8c), whilst DEGs related to protein ubiquitination and acetylation, protein nuclear transport, and heterochromatin formation and transcriptional regulation were enriched in eWAT from cold-exposed obese mice treated with KPT-330 (Supplementary Fig. 8h). Whether skeletal muscles and WAT represent key loci for supporting cold survival, and how FOXO1 contributes to the process in these organs remain to be elucidated.

Our current findings are limited by the fact that the transport of other nuclear proteins is likely also interfered by KPT-330. Hence, though we did not observe overt adversary consequences in our studies, caution should be made towards the dosage of this anti-cancer drug. Future drug candidates specifically and reversibly targeting FOXO1 SIM or SUMOylation sites on IPO7 or XPO1 would be preferred.

## Methods

The use of human tissue samples complied with the Declaration of Helsinki and was approved by Institutional Ethics Committee of the Third Affiliated Hospital of Sun Yat-Sen University (No. 02-058-01). All animal studies were in strict accordance with the recommendations in the Guide for the Care and Use of Laboratory Animals of the National Institutes of Health. All animals were handled according to approved institutional animal care and use committee (IACUC) protocols of the Third Affiliated Hospital of Sun Yat-sen University and China Zebrafish Resource Center.

### Human donor pancreatic tissues

Gender of pancreatic donors who had provided informed consent was listed in Supplementary Table 1. The relatives or guardians of donors in this study gave written informed consent. The median age, body weight and body mass index of donors included in this study were 42 years old, 60 kg, and 22.9 kg/m$^2$. These donors had no known history of diabetes.

Mediums and reagents used are in Supplementary Table 2. Human pancreatic tissue was obtained from deceased patients consented to donate their organs for research. Small pieces of donor pancreatic tissues were placed into University of Wisconsin solution (UW) or

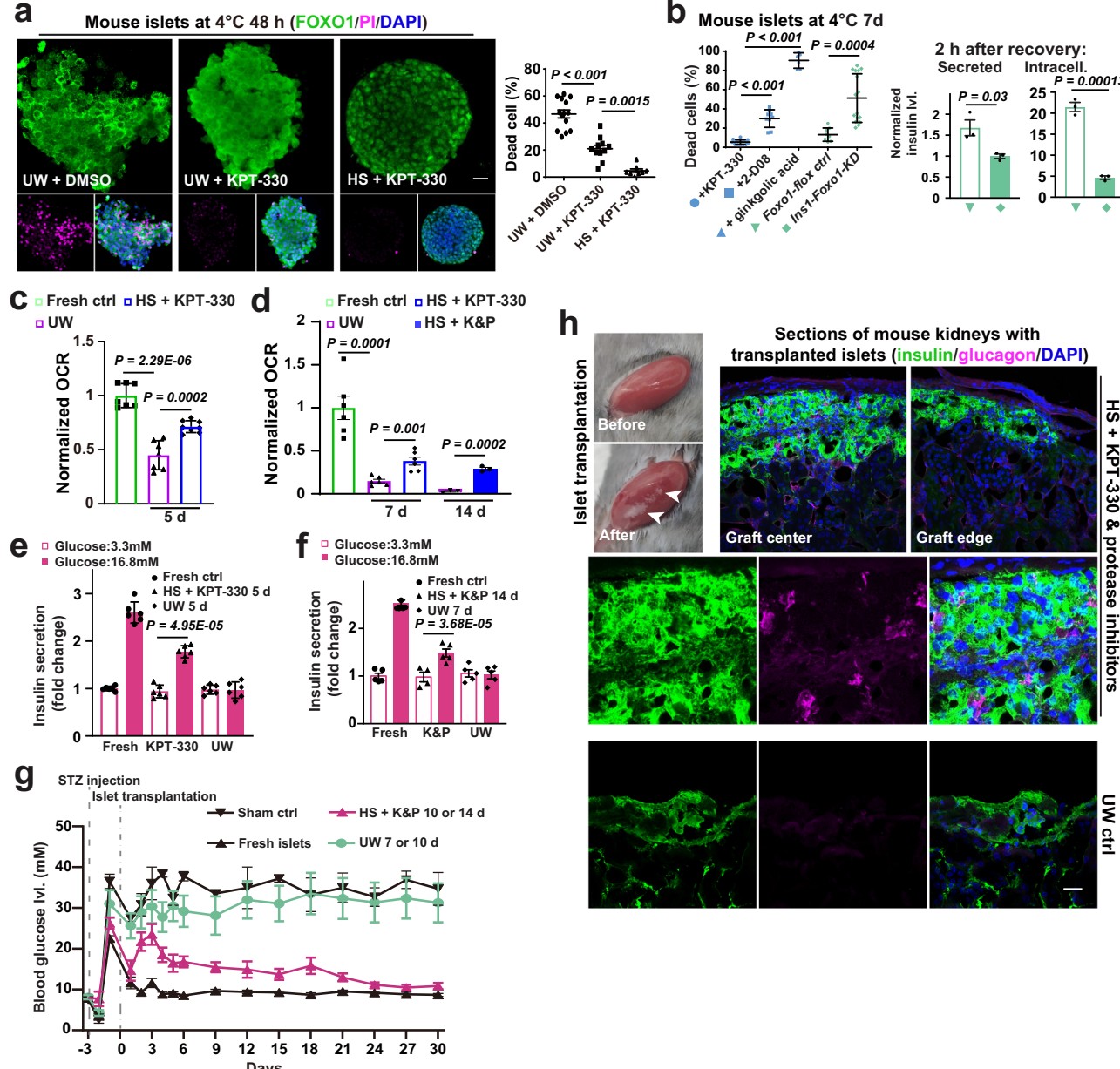

**Fig. 7 | Mouse pancreatic islet long-term storage and transplantation. a** Left: confocal images of FOXO1, PI and DAPI staining in mouse islets at indicated conditions; right: statistics on cell death percentage (from left to right, $n$ = 14, 11 and 8 images from 3 experiments); UW, University of Wisconsin Solution; HS, basal hibernation solution used in cell cold exposure experiments. **b** Left: dead cell percentage in cold-stored mouse islets at annotated conditions (from left to right, $n$ = 14, 9, 6, 8 and 16 images from 3 experiments); 2-D08 and ginkgolic acid are SUMOylation inhibitors added to HS; *Ins1-Foxo1-KD*, pancreatic β cell-specific knock-down of *Foxo1*; right: normalized levels of secreted and intracellular insulin in rewarmed islets as indicated ($n$ = 3 experiments). Normalized islet oxygen consumption rate (OCR) from indicated conditions, as an assessment of islet quality following 24-h (fresh and UW: $n$ = 7 experiments; KPT-330: $n$ = 8 experiments) (**c**) or 2-h ($n$ = 6 and 3 experiments for 7 and 14 d, respectively) (**d**) rewarming in vitro;

K&P, KPT-330 and protease inhibitors. Glucose-stimulated insulin secretion assays with mouse islets from indicated conditions following 24-h ($n$ = 6 experiments) (**e**) or 2-h ($n$ = 5 experiments) (**f**) rewarming in vitro. **g** Blood glucose levels in Streptozotocin (STZ)-treated diabetic mice with cold-stored islets at indicated conditions (mice with sham transplantation: $n$ = 3; transplanted with fresh islets: $n$ = 5; $n$ = 3 each for UW 7 or 10 d; $n$ = 5 and 6 for HS + K&P 10 d and 14 d, respectively). **h** Representative image showing the transplanted islets (arrowheads), and confocal images of insulin, Glucagon and DAPI staining on kidney sections of the diabetic mice transplanted with islets stored at 4 °C in HS + K&P for 14 days ($n$ = 6 mice), or with islets stored in UW solution at 4 °C for 5 days ($n$ = 3 mice). Data are shown as mean and SEM. Statistics: one-way ANOVA followed by Tukey's test (**a**, **b**) and two-tailed Student's $t$ test (**b**–**f**). Scale bars: 20 µm (**a**, **f**). Source data are provided as a Source Data file.

'hibernation solution' (HS) for various duration at 4 °C. For insulin measurement, the cold-stored pancreatic tissues were cut into smaller pieces (50 mg each), added 800 µl of ice-cold protein extraction reagent and homogenized. The lysate was centrifuged at 16,000 g for 10 min, the supernatant was collected, and insulin was measured using an ELISA kit.

## Animals

The C57BL/6 (Gempharmatech, Nanjing, China), *Ins1-IRES-iCreERT2* (Biocytogen, Beijing, China) and *Foxo1-P2A-EGFP-flox* (Gempharmatech, Nanjing, China) mice were bred at the animal facility of the Third Affiliated Hospital of Sun Yat-sen University in accordance with the requirements of the institutional Animal Review Board. Both male and

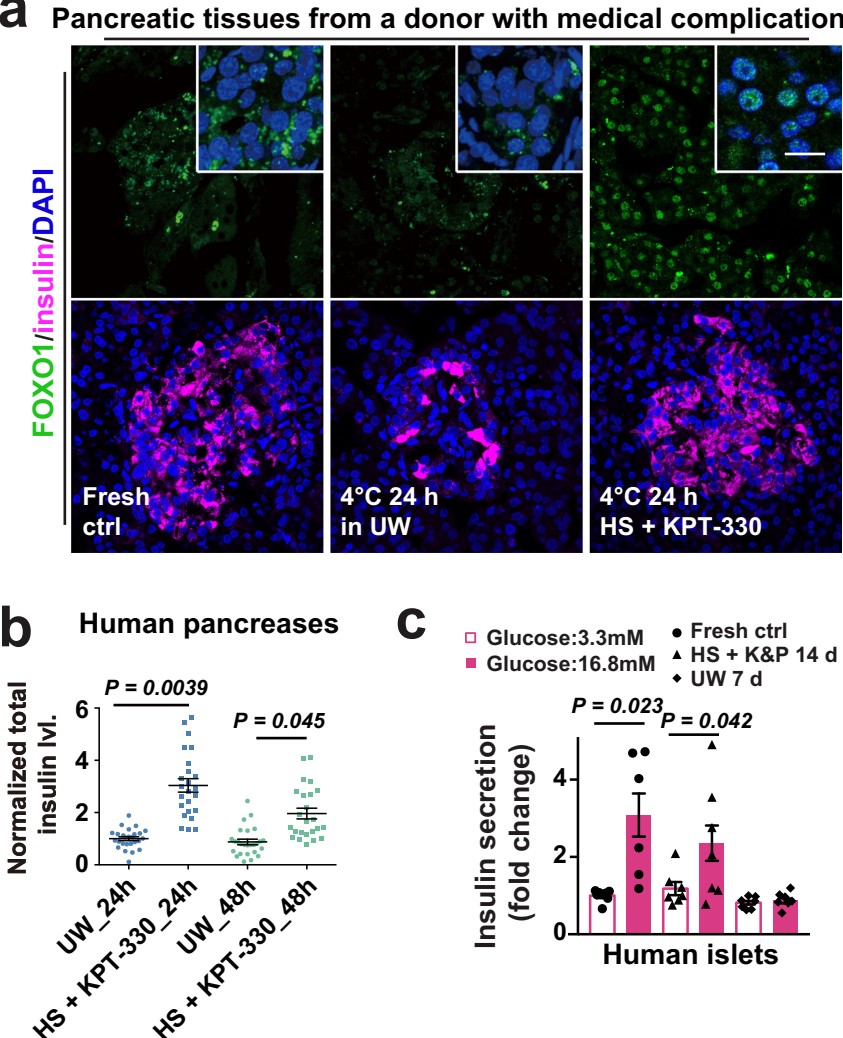

**a Pancreatic tissues from a donor with medical complications**

**Fig. 8 | Static cold storage of human pancreatic tissues and islets. a** Confocal images of FOXO1, insulin and DAPI staining in human pancreatic sections at indicated conditions (*n* = 9 donors). **b** Normalized levels of total insulin in human pancreatic tissues at annotated cold storage conditions (*n* = 3 samples from each of 9 donors). **c** GSIS assays with islets from human at indicated conditions (fresh ctrl: *n* = 6 donors; UW and K&P: *n* = 7 donors, respectively); K&P, KPT-330 and protease inhibitors. Data are shown as mean and SEM. Statistics: two-tailed Student's *t* test (**b**, **c**). Scale bars: 20 μm (**a**). Source data are provided as a Source Data file.

female mice were used in preliminary experiments, and no gender-specific difference was noted. For collection of pancreases and pancreatic islets, and islet transplantation, male mice of 12–16 weeks old were used. For obese mice (12–14 month) (Gempharmatech, Nanjing, China), mice were switched to high-fat diet (OpenSource Diets, D12492) with 60% kcal from fat, 20% kcal from carbohydrate, and 20% kcal from protein for 3 months. The gender of mice used in cold exposure experiments is provided in Data Source file. All mice were housed with a 12/12 light/dark cycle at 20–25 °C and 45–65% of relative humidity.

Zebrafish (Danio rerio) lines AB-wild type and Tg(myl7:EGFP) were acquired from China Zebrafish Resource Center and raised at 28 ± 0.5 °C in 14 h light/10 h dark cycles. As zebrafish larvae were used in this study, their gender was not identified, whilst presumably about equal amounts of male and female fish were used in each experiment.

**Cell lines**

Culture mediums and reagents are listed in Supplementary Table 2. TLGS iPSCs and iPSC-neurons were established and maintained as described previously[9,10]. TLGS iPSC colonies were cultured in Geltrex or 4% Matrigel-coated dishes in TLGS iPSC medium. For TLGS iPSC-neuronal differentiation, TLGS iPSC cultures were grown to 80–90% confluence, dissociated by accutase, transferred to 96-well plates at about 5000 cells per well and cultured as embryoid bodies (EBs) for 3–5 days in NPC medium. The EBs were transferred to 6-well plates at about 20 EBs per well and cultured for 3–5 days in ND1 medium. Then, the EBs were digested by trypsin/EDTA, dissociated cells were plated onto poly-L-ornithin/laminin -coated dishes at about 300,000 cells per 35-mm dish, cultured in ND1 medium for 2 days, ND2 medium for 2 days, and maintained in ND3 medium for up to 14 days.

Human iPSC-neurons were cultured in the W.LI. lab at the National Eye Institute following the NIH stem cell protocols (https://stemcells.nih.gov/research/nihresearch/scunit/protocols.htm).

H1 ESC line was originally obtained from WiCell (Madison, USA), cultured in matrigel-coated plates in mTeSR medium and passaged following the standard protocol. The usage of H1 ESC has been in accordance with guidelines and regulations by Institutional Ethics Committee of the Third Affiliated Hospital of Sun Yat-Sen University.

Human retinal epithelial ARPE-19 cell line was obtained from the American Type Culture Collection (ATCC, Manassas, VA) and maintained in standard medium.

All human cell lines were authenticated by STR analyses by a local service company every 6 months.

## Cold exposure experiments

Cold exposure experiments were performed similar to that described previously[9]. Briefly, cells cultured at normal conditions were used as 37 °C controls. In drug treatment groups, DMSO was added into the control cultures, and we consistently observed that DMSO (<0.1%) would weaken the cold-induced FOXO1 nuclear entry (for example, see Fig. 1e), but did not significantly impact our experiments and conclusions. Drugs (Supplementary Table 2) were first added to the cultures and incubated at normal conditions for 30–60 min. Then the DMSO or drug-containing culture mediums were replaced with the Hibernate-A medium (Supplementary Table 2) supplemented with the same types and amounts of supplements used in their normal mediums and drugs. For example, for ARPE-19, 10% fetal bovine serum and 1x antibiotic antimycotic were added into Hibernate-A; for H1 ESCs and TLGS iPSCs, the same amounts of growth factors and other supplements were added into Hibernate-A; if 10 μM FOXO1 or other drugs were incubated with the cells at 37 °C in their normal mediums, then the cells were also incubated with the same amount of the drug in Hibernate-A. The cultures were then incubated at room temperature for 15 min, and then transferred to a 4 °C refrigerator for the designated time. For the rewarmed groups, the cultures were removed from the refrigerator and allowed to rewarm to room temperature for 15 min; the Hibernate-A medium was replaced with normal culture mediums, and the cells were cultured at normal conditions for the designated time.

For zebrafish, zebrafish larvae at 6, 12, 24, 48, 72 and 120 h post fertilization (hpf) were first subject to gradient cooling to 4 °C in 30 min in a PCR machine, then transferred into a 4 °C refrigerator for 4 or 10 h, followed by rewarming to 28 °C for 2 h and microscopic imaging on larva survival and morphology. Abnormal or dead larvae were removed, and the rest of the larvae were allowed to grow to adulthood with weekly inspections for any apparent developmental defects. Zebrafish embryos at 72 hpf consistently demonstrated good cold tolerance. Hence, 25 ng of morpholino-modified control or zebrafish *foxo1a* morpholino antisense oligonucleotides (Supplementary Table 3) were microinjected into fertilized zebrafish eggs (0 hpf). These injected eggs were then allowed to develop till 72 hpf for cold-exposure experiments.

## Sample preparation for ATAC sequencing

ATAC-seq was done with the NovoNGS® Chromatin Profile Kit for Illumina (N248 Novoprotein, Shanghai, China). In brief, a total of 50,000 H1 cells incubated at 37 °C or 4 °C were washed twice with 50 μl of PBS at 37 °C or 4 °C, respectively. The cells were resuspended in 50 μl lysis buffer (10 mM Tris-HCl, 10 mM NaCl, 3 mM MgCl$_2$, 0.5% NP40, pH7.4). The suspension of H1 cell nuclei was then centrifuged for 5 min at 500 g at 4 °C, followed by the addition of 40 μl Tn5 transposase reaction mix (20 μl 2x TD buffer, 0.4 μl 1% Digitonin, 0.4 μl 10% Tween 20, 4 μl 3×PBS, 13.2 μl nuclease-free H$_2$O and 2 μl Transposome). The transposition reaction was incubated at 37 °C for 30 min. Equimolar Adapter1 and Adapter 2 were added after transposition. PCR was then performed to amplify the library. After the PCR reaction, libraries were purified with the AMPure beads and library quality was assessed with Qubit. The clustering of the index-coded samples was performed on a cBot Cluster Generation System using TruSeq PE Cluster Kit v3-cBot-HS (Illumina) according to the manufacturer's instructions. After cluster generation, the library preparations were sequenced on an Illumina Hiseq platform and 150 bp paired-end reads were generated (Novogene, Beijing, China).

## Sample preparation for CUT&Tag sequencing

CUT&Tag was done with the NovoNGS® CUT&Tag High-Sensitivity Kit (N259; Novoprotein, Shanghai, China). Briefly, H1 cells incubated at 37 °C or 4 °C were harvested, counted, split into 200,000 cells per group and centrifuged for 4 min at 500 g at 37 °C or 4 °C, respectively. The cells were lysed (Supplementary Table 2) by gentle pipetting and placed on ice for 10 min. The cell nuclei remained intact by this treatment, and were collected by centrifuged for 5 min at 500 g at 4 °C. The nucleus fraction were resuspended by 200 μl PBS, cross-linked by adding 2 μl of 10% Formaldehyde and incubated at room temperature for 2 min. The cross-link reaction was terminated by adding 5 μl of 25 mM glycine. The cross-linked cell nuclei were centrifuged for 5 min at 600 g at room temperature. The pellets were resuspended in 90 μl wash buffer. Then, 10 μl of Concanavalin A-coated magnetic beads were equilibrated at room temperature and added into each sample, incubation continued for 15 min. The unbound supernatant was removed and bead-bound cells were resuspended in a 1:50 dilution of FOXO1 primary antibody (Supplementary Table 4). Primary antibody incubation was performed on a rotating platform at 4 °C overnight. Followed by incubation with a secondary antibody at room temperature for 1 h to increase the number of IgG molecules at each epitope bound by the primary antibody. Cell nuclei were washed and incubated with pA-Tn5 adapter complex at room temperature for 1 h and washed under stringent conditions following manufacturer's instruction. Next, cell nuclei were resuspended in 50 μl Tagmentation buffer by addition of Mg$^{2+}$ and incubated at 37 °C for 1 h. To stop tagmentation, 1 μl 10% SDS was added to 50 μl of sample, which was incubated overnight at 37 °C. To extract the genomic DNA, Tagment DNA Extract Beads were added to each tube. Samples were enriched by PCR amplification and a single solid-phase reversible immobilization (SPRI) magnetic bead cleanup step. For the negative control, we omitted the primary antibody, while the secondary antibody could randomly coat the chromatin at low efficiency without sequence bias. The purified DNA was used for library preparation. Subsequently, library quality was assessed on the Agilent Bioanalyzer 2100 system, and pair-end sequencing was performed on an Illumina platform (Novogene, Beijing, China).

## Processing and analyses of ATAC-seq and CUT&Tag data

Fastp v0.21.0[59] was used to remove adapter and low-quality reads. Paired end reads were then mapped to human genome (Ensembl GRCh38)[60] using Bowtie2 v2.4.2[61]. After alignment, unique aligned reads were filtered with map quality and duplicates (Picard MarkDuplicates v2.25.0; http://broadinstitute.github.io/picard/). For ATAC-seq analysis, reads mapped to MT chromosome and reads overlapping the ENCODE hg38 functional genomics regions blacklist[62] were also removed to improve the quality of the retained fragments. To correct for the fact that the Tn5 transposase binds as a dimer and inserts two adapters in the Tn5 tagmentation step, all positive-strand reads were shifted 4 bp downstream and all negative-strand reads were shifted 5 bp upstream to center the reads on the transposase binding event. Normalized, fragment signal bigwigs were created using Deeptools v3.5.1[63] (bamCoverage; RPGC normalization), and visualization was performed with Integrative Genomics Viewer (IGV) software[64]. Peaks were then called individually for each replicate using macs2 v2.2.7.1[65] (options: -f BAMPE --call-summits), with $q = 0.05$ as the threshold for ATAC-seq peaks and $q = 0.1$ for CUT&Tag peaks. Heatmap and peak annotation were performed with ChIPseeker v1.5.1[66] and human genome (Ensembl GRCh38)[60]. Differential analysis of peaks in ATAC-seq was performed using EdgeR embedded in Diffbind v2.14.0[67] (option: summits = 200), and differential analysis of FOXO1-binding peaks in CUT&TAG were performed using DESeq2 embedded in Diffbind v2.14.0[67]. Peaks with an FDR-corrected Q-value < 0.05 were assigned as differentially expressed. Pathway analysis were performed using the Metascape[68] web interface. Motif enrichment of differential binding regions (FDR < 0.05;|Fold | > 1.5) in ATAC-seq was performed with Homer v4.11.1[69](findMotifsGenome.pl; -size given). Top motifs were reported and compared to the Homer

database of known motifs. The motifs present in targets sequences less than 5% were removed.

## RNA-seq sample preparation

H1 ESC line was maintained and prepared as described in this work. Experiments were conducted by first replacing H1 culture medium with fresh Hibernate-A based medium, allowing the cells to sit at room temperature for 30 min. The samples were divided into these groups: 37 °C ctrl, 4 °C 4 h, and 4 °C 4 h_37 °C 2 h (rewarmed). Cells were collected and lysed in TRIzol (Supplementary Table 2), total RNA were purified with RNeasy® Mini kit (74104; Qiagen, Hilden, Germany), and further processed for mRNA purification using poly-T oligo-attached magnetic beads, and RNA-sequencing (Novogene, Beijing, China).

Obese mice (12–14 month) were divided into ambient temperature control, DMSO_cold exposed, and KPT-330_cold exposed groups. Mice subjected to KPT-330 or DMSO injection as described above were housed at 4 °C for 6 h. Samples of the epididymal white adipose tissues (eWAT), brown adipose tissues (BAT), skeletal muscles and liver from the 3 groups of mice were quickly collected and lysed in TRIzol. Subsequent RNA preparation and sequencing are as described above.

## Quality control, mapping and quantification of RNA-seq data

Clean reads were obtained by removing adapter, and excluded reads containing ploy-N and low-quality from raw data. Reads were aligned to the mouse genome (Ensemvl GRCm38) or human genome (Ensembl GRCh38)[60] with Hisat2 v2.0.5[70], and only the reads aligned to exons, and with the mapping quality score larger than 10 were counted using the featureCounts v2.0.1[71]. Given the pair-wise design of the experiments, differentially expressed genes were identified using DESeq2 v1.26.0[72] by control of the effect of different cell passages. $P$ values from differential expression tests were adjusted using the Benjamini-Hochberg procedure for multiple hypothesis testing, and genes with an adjusted $P < 0.05$ were identified as differentially expressed. Pathway analysis were performed using the Metascape[68] web interface.

## CRISPR/Cas9 editing of the FOXO1 SIM locus in ARPE-19 cells

ARPE-19(FOXO1-minus SIM) cell line was established with the Clustered regularly interspaced short palindromic repeats-(CRISPR)-Cas9 system. SgRNAs for CRISPR/Cas9 editing were designed, synthesized (Generay Biotech., Shanghai, China), and cloned into a lentiviral Cas9/sgRNA vector LV-U6-spsgRNA1-CMV-SV40-NLS-hspCas9-NLS-Flag-P2A-Puro-T2A-EGFP-WPRE. Editing efficiency of each sgRNA was tested in HEK293T cells using T7 Endonuclease I (EN303-01/02, vazyme) according to manufacturer's instructions.

To seamlessly delete the predicted SIM at FOXO1 gene locus in ARPE19 cells, $1.2 \times 10^6$ ARPE19 cells were plated in P6-well plates and cultured overnight. Then the cells were transfected with 2 μg of Cas9/sgRNA plasmids using FUGENE HD (Promega, Cat# E2311). 24-h after transfection, cells were selected with 4 μg/ml puromycin for 48 h. 24-h after selection, cells were dissociated into single cells, serially diluted to a final concentration of 10 cells per 1000 μl medium, and 100 μl of diluted cells were added to each well of a 96-well plate. The medium was replaced every 5 days and colonies were inspected for a clonal appearance ~1 week after plating. Only colonies from single cells were passaged and characterized. Genomic DNA was extracted, and fragments flanking sgRNA-targeting regions were amplified (815 bp) and sequenced with the following primers (Forward 5′- GATGGCCCCGC-GAAGTTAAGTT-3′; Reverse 5′- CGAGCTGTTGCTGTCACCCTTA −3′). One colony with FOXO1SIMdelins mutation was confirmed with western blot and used for further experiments.

## Proximity ligation assay

PLA was utilized to detect in situ protein−protein. Before starting, the samples should be deposited on glass slides and pre-treated with respect to fixation and permeabilization. The samples were then processed using a PLA kit (DUO92101, Sigma-Aldrich) according to the manufacturer's instructions. A ZeissLSM880 confocal microscope detected PLA signals as discrete punctate foci and localized complexes intracellularly. The total intensity of the PLA signal per cell was quantified using Fiji software.

## Transfection of overexpression plasmids and siRNA

Plasmids, primers and siRNAs used are listed in Supplementary Table 3. Cell transfection reagents used are listed in Supplementary Table 2. Plasmids were transfected into the cells with Lipofectamine 3000 following manufacturer's instruction. After 12 h at 37 °C the medium was replaced with fresh culture mediums for each cell types. The cells were further cultured for 24−48 h post-transfection before cold exposure. Cold exposure experiments were performed as described above.

## Quantitative PCR

Primers used for real-time PCR are shown in Supplementary Table 3. Total RNA was extracted and purified using RNeasy® Mini kit (74104; Qiagen, Hilden, Germany) according to the manufacturer's instruction. Total RNA concentration was determined by NanodropTM Spectrophotometer 2000 (Thermo Fisher, USA). Then, 1.5 μg of RNA from each sample were reverse transcribed with high-capacity cDNA reverse transcription kit (Roche, Germany). Real-time PCR was performed using the SYBR Green PCR Master Mix (Roche, Germany) in the 7900HT Fast Real-Time PCR system (Roche, Germany).

## Western blotting, co-immunoprecipitation and immunostaining

All primary antibodies are listed in Supplementary Table 4. Mammalian protein extraction buffer (Supplementary Table 2) was used to lyse the cells and extract proteins. Total proteins or proteins in the soluble fraction after standard centrifugation removing precipitates and debris were run on SDS-PAGE. Standard protocols were used for western blotting, co-immunoprecipitation and immunostaining. Western blots were quantified with the ImageJ Gel Analysis tool.

## Confocal microscopy, STED microscopy and image analysis

All immunostaining images were captured on a Zeiss LSM 880 confocal system (Zeiss, Oberkochen, Germany) or by STED imaging (Leica, Wetzlar, Germany). STED images were acquired with Leica TCS SP8 STED 3 x microscope that is equipped with a 100 ×1.4 NA HC PL APO CS2 oil immersion objective and operated with the LAS-X imaging software. Excitation was with a tunable white light laser and emission was detected with hybrid detectors. In time-gated STED mode, DAPI, Alexa 594 and Atto 647 N were sequentially excited at 405, 580 and 640 nm, respectively, with the 775 nm STED depletion beam, and their fluorescence collected at 415−570 nm, 585−635 nm and 645−760 nm, respectively. The STED resolution in our conditions was ~60 nm on the microscopic X- and y axis (parallel to the coverslip), and ~150−200 nm on the microscopic Z axis. STED images were deconvolved using Huygens software (Scientific Volume Imaging).

To analyze cellular distribution of proteins of interest, we quantified their fluorescence intensity in nucleus, cell cytosol and the whole cell with Image J. The boundary between nucleus and cytosol can be well visualized with DAPI co-staining. The $F_{FOXO1}$ ratio between nucleus and cytosol (Nuc/Cyto), and whole cell (Nuc/Cell) were calculated for each cell individually before further analysis. To reduce variation caused by cell size and photobleaching, for images acquired in XYZ scanning, only the cells with its maximum diameter and minimum photobleaching were used for further analysis.

## Glucose tolerance test (GTT)

Mice were deprived of food for 16 h. the basal blood glucose levels were determined by tail bleeding using a blood glucose glucometer (Accu-Check, Roche). The mice were intraperitoneally injected with a

bolus dose of glucose (1.5 mg of glucose per g of body weight). Blood glucose levels were then measured at the times indicated.

### Indirect calorimetry

Heat, $VO_2$, $VCO_2$, food intake and activity movement of obese mice (12–14 months) and young healthy mice (8–10 weeks) were monitored using the Comprehensive Laboratory Animal Monitoring System (CLAMS, Columbus Instruments). Before starting, mice were acclimatized to 22 °C for 4 days in metabolic cages. Over the course of the experiment the temperature was adjusted as depicted in the graphs ranging from 22 °C to 4 °C. For KPT-330 treatment, mice were injected i.p. with KPT-330 (Selleck, 3 mg/kg) 30 min prior to acute cold stimulation; for vehicle control, mice were i.p. injected with equal amount of DMSO (1 ml/kg). Mice were monitored in the calorimetry chambers with recordings at 4-min intervals. Obtained indirect calorimetry data were analyzed using Prism7.0.

### Body composition analysis

The fat and lean mass of the mice were analyzed using a 3-in-1 Echo MRI composition analyser (Echo Medical Systems, 100H).

### Islet isolation, cold storage and functional assessments

Mediums and reagents used are listed in Supplementary Table 2. Human pancreatic islets were isolated as described[73]. The pancreatic samples were enclosed in an isolator, which consists of two stainless steel chambers with a mesh in the middle. Glass marbles and collagenase P solution were added to the chamber during the digestion process. The chamber was connected to a vibrator, and the digested pancreatic samples spontaneously passed through the mesh. The pass-through fraction was then cooled to room temperature to terminate the digestion. This fraction was further filtered through a smaller mesh, and the islets were purified by handpicking. The handpicked islets were then resuspended in RPMI 1640 medium. Human islets can be maintained in RPMI 1640 at 37 °C for up to 4 days.

Mouse pancreatic islets were removed and isolated by collagenase P digestion as previously described with minor modifications[74]. Briefly, C57BL/6 J mice of 8–14 weeks old were sacrificed by cervical dislocation and sprayed with 70% ethanol. The pancreas and common bile duct were exposed. The pancreas was inflated by cold collagenase P solution through the common bile duct with a 30 g needle. Then the pancreas was removed and placed in a 50 ml tube containing collagenaseP solution and incubated in a 37 °C water-bath for 12 min with vigorous shaking. The digestion was terminated by adding 25 ml of ice-cold RPMI 1640 medium. The tube was centrifuged at 1000 rpm for 2 min, supernatant discarded. Then the pellet was resuspended with 20 ml ice-cold RPMI 1640 medium and filtered through a 0.4–1.5 mm wire mesh. The filtered fraction was again centrifuged at 1000 rpm for 2 min, pellet gently resuspended with 10 ml histopaque 1077. Another 10 ml RPMI 1640 medium was added around the inside of the tube and centrifuge the tube at 2400 rpm for 20 min with very slow acceleration and braking. The isolated islets were carefully collected from the medium interface and transferred to a tube containing 50 ml RPMI 1640 medium. The tube was centrifuged at 1200 rpm for 3 min, supernatant discarded. The islets were resuspended with 2 ml RPMI 1640 medium and placed into a 35 mm dish, hand-picked using a pipette with a wide-open tip. The islets were counted and transferred into RPMI 1640 medium and incubated in a 5% $CO_2$ incubator at 37 °C. Mouse islets can be maintained at this condition for up to 5 days.

As described previously[9], hibernation solution (HS) and protease inhibitors mitigate cold-induced injury in mammalian cells and organs. Thus, here we used the UW solution as the standard control, HS was used as a novel basal islet storage solution, and protease inhibitors were added for extra protection for storage > 7 days. For cold storage of human and mouse islets, 100 donor islets or 30 mouse islets of 100-

200 μm diameter were transferred to a 24-well plate with 1 ml of UW or HS in each well. The islets were allowed to cooled to ambient temperature for 30 min, then transferred to a 4 °C refrigerator for designated days.

Following cold storage, for glucose stimulated insulin secretion, the islets were gradually rewarmed to room temperature on the bench for 15 min, transferred to RPMI 1640 medium and incubated at 37 °C for 4 h. Then the islets were transferred to a 24-well plate with 500 μl Krebs–Ringer bicarbonate (KRB) buffer and starved at 37 °C for 1 h. The islets were then incubated with low glucose (3.3 mM) KRB buffer at 37 °C for 1 h, and 200 μl of the buffer was carefully collected for later ELISA assay. Then the islets were incubated with high glucose (16.7 mM) KRB buffer at 37 °C for 1 h, and 200 μl of the buffer was carefully collected for later ELISA assay. The remaining islets were lysed in 30 μl protein extraction buffer. The collected KRB buffer samples and lysed islets were subjected to insulin ELISA assay. Islet oxygen consumption rate (OCR) and mitochondrial DNA/nuclear DNA measurements were performed as previously described[44,45], following manufacturer's instruction (ab197243, Abcam, Cambridge, UK). OCR signals were recorded at 2 min intervals for 60 min at Excitation/Emission = 380/650 nm. OCR data were corrected by linear regression, and normalized to the concentration of islet total DNA that was measured following the OCR assays. Islet mitochondrial content was determined by the ratio of mitochondrial DNA/nuclear DNA as described previously[75].

### Mouse islet transplantation

To induce diabetes, C57BL/6J mice of 8–14 weeks old were injected a dose (150 mg/kg body weight) of streptozotocin (STZ, Sigma-Aldrich) dissolved in sterile citrate buffer (50 mg/ml) intraperitoneally. Mice were fed with 20% dextrose water on the first day after injection to avoid hypoglycemia. Following the surgery, mice were then fed with normal diet. Blood glucose was sampled by puncturing tail vein. Mice with blood glucose levels over 20 mM for two consecutive days were identified as diabetic. Transplantation procedure of mouse islets was performed as described[76]. Briefly, mouse was anesthetized by inhalation of isoflurane and placed in prone position. The left lumbar region was shaved and disinfected with 75% ethanol. A 2 cm incision was made and left kidney was exposed through the peritoneal opening. A 0.2 cm incision was made on the kidney capsule to create a subcapsular pouch. Using a PE50 tube, 300 cold-stored islets were transplanted into the subcapsular space of the left kidney. Surgery wounds were carefully sutured, and bupivacaine hydrocholoride was administered to relieve pain. Mice were closely monitored; blood glucose levels were measured every day for the first week and then every 3rd day until 30 days after transplantation. Cure of diabetes was defined as blood glucose level less than 11.1 mM in two consecutive days or more. At the end of the experiment, islets-bearing kidney was removed, fixed in 4% paraformaldehyde, and processed for immunofluorescence.

### Statistics and reproducibility

We set sample sizes for experiments based on preliminary experiments. Representative results were from at least 3 independent experiments with similar outcomes. For numerical variables, all data were reported as the mean and SEM. $P < 0.05$ was considered statistically significant and two-tailed Student's $t$ test was used to compare the differences between two groups, whilst one-way ANOVA followed by Tukey's test was used to assess the differences among multiple groups. For all experiments, all stated replicates were biological replicates. For in vivo studies, mice were randomly assigned to treatment groups. GraphPad Prism was used as the primary statistical software in the study. To obtain the exact $P$ values from Student's $t$ test, data were also analyzed in Microsoft Excel.

**Reporting summary**

Further information on research design is available in the Nature Portfolio Reporting Summary linked to this article.

## Data availability

All sequencing data are available at NCBI GEO under accession number GSE185152. This paper does not report original code. Source data are provided with this paper.

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

## Acknowledgements

We thank Lue Xiang, Haishan Zhao for their help in the zebrafish cold-adaptation experiments, Xiaoyi Xu for her advice in the ARPE-19 CRISPR experiment and Jinliang Liang, Xiaofen Zhong for helping the RNA-seq experiments. Funding was provided by the Intramural Research Programs of the National Eye Institute (WLI), the National Natural Science Foundation of China (82170671 to J.O., 82172585 to WLIU, 81770648 and 81972286 to Y.Y.); the National Key R&D Plan (2017YFA0104304 to Y.Y.); the Natural Science Foundation of Guangdong Province (2015A030312013 to Y.Y., 2018A030313259 to WLIU); Science and Technology Program of Guangdong Province (2019B020236003 and 2020B1212060019 to Y.Y.); Guangdong Basic and Applied Basic Research Foundation (2021A1515010519 to T.D.).

## Author contributions

X.Z., W.Li and J.O designed experiments; X.Z., L.G., G.J., Q.Yu., L.C., W.C., T.D., J.S., J.Y., J.O. performed experiments and analyses; Y.L. performed CUT&Tag, ATAC and Transcriptomic data analyses; G.L., Y.X., Q.Y., L.Y., S.Y., H.L., Q.Z., G.C., W.Liu. and Y.Y. provided donor pancreatic samples and cell lines; X.Z., L.G., Q.Yu., J.K.M., W.Li. and J.O. supervised the study and cowrote the paper; T.D., G.C., W.Liu., Y.Y., W.Li. and J.O. obtained the funding.

## Competing interests

The authors declare no competing financial or non-financial interests. A Patent Corporation Treaty PCT application (PCT/US2021/064086), with NIH and SYSU inventors was filed by the NIH on December 17th, 2021 that claims priority to the Chinese patent (CN112602703B) for Preparation Method and Application of Cell, Tissue and organ cold preservation liquid.
