## [Peer Review File · Nature Communications]

Cold-induced FOXO1 nuclear transport aids cold survival and tissue storageREVIEWER COMMENTS

Reviewer #1 (Remarks to the Author):

In this manuscript Zhang et al. aim to reveal the molecular mechanisms being relevant for cold-induced translocation of FOXO1. In my review, and given my field of expertise, I focus on the structural aspects of the manuscript. After having tested FOXO1 phosphorylation and acetylation in order to explain its nuclear retention, the authors aim to come up with alternative explanations. Overall, in its present form, the line of argumentation is difficult to follow and the statements are far-fetched in my opinion. To me it seems that the bottom line of this part of the manuscript is that XPO1/CRM1 and IPO7 are important for re-localization of FOXO1. However, the role of SUMOylation or why the authors need SUMOylation at all to explain FOXO1 translocation is entirely unclear to me. Furthermore, it is unclear how the authors came up with the SUMO-interacting motifs for FOXO1 as the motif is an extremely short sequence and the manuscript lacks a convincing explanation for how this binding site was found and how it was determined. In line with this, the MD results and their interpretation seem far-fetched. Assuming XPO1/CRM1 and IPO7 as the key factors, and given that the NLS/NES of FOXO1 has already been identified, I'm wondering why the authors didn't mutate the FOXO1 NLS/NES and why they didn't use XPO1/CRM1 and IPO7 inhibitors to validate the interaction. For FOXO1 it has already been reported that phosphorylation of several residues by PKB/AKT, CK1 and DYRK can lead to nuclear exclusion. Not all of these sites have been evaluated by the authors. Seeing that that the DYRK site seems to become dephosphorylated after 4h at 4°C (SFig. 3A), I suggest that, next to the inhibitors, the authors test relocalization of P-mimicking (Asp or Glu) and P-deficient (Ala) FOXO1 mutants.

Reviewer #2 (Remarks to the Author):

The manuscript clearly revealed the involvement of FOXO1 in regulating cold stress in human and mouse cells. The authors reported FOXO1 translocation from cytoplasm to nucleus with SUMO-modification to be the key to relay the cold signal and elicit the cold responses. A nonconventional, temperature-sensitive FOXO1 transport mechanism involving the nuclear pore complex protein RANBP2, Importin-7 and Exportin-1, and a SUMO-interacting motif on FOXO1 was found to control nuclear entry of FOXO1. More importantly, the authors found the Exportin-1 inhibitor KPT-330 greatly enhanced cold tolerance prolonged the shelf-life of human and mouse pancreases and islets. These findings are original and extending current understanding on mechanisms of cellular cold tolerance. The results have potential to promote cold therapy and cold perservation of human tissues as the authors have demonstrated in the paper.

The work is well executed and evidence is adequate to support the authors' claims. The methodology used are in the cutting edge of the field.

I do have a small concern that the authors might be able to clarify further:

1. Through Cut&Tag and RNA-seq integration analysis, the authors found that the Foxo1 binding motif was present in 20% of genes whose expression was significantly changed under cold treatment. However, I have a concern about the role of Foxo1 as the major contributor. Is it possible that other Foxo family members or transcription factors also participate in the regulation of the remaining 80% of genes? In this manuscript, the authors wrote: "Also, FOXO1 proteins were enriched in TLGS cell nuclei of multiple tissues and organs during hibernation or following storage at 4°C, while FOXO3 did not show such dynamics, suggesting a specific role of FOXO1 in cold adaptation during hibernation (Supplementary Fig. 7e-g)." No dynamic pattern as FOXO1 does not preclude the role of FOXO3 in cold response, as some literature indicated such involvement. Hu P, Liu M, Liu Y, Wang J, Zhang D, Niu H, Jiang S, Wang J, Zhang D, Han B, Xu Q, Chen L. Transcriptome comparison reveals a genetic network regulating the lower temperature limit in fish. *Sci Rep.* 2016 Jun 30;6:28952. doi: 10.1038/srep28952. AND other studies reveal the involvement of FOXOs in thermal responses. (Dettleff P, Zuloaga R, Fuentes M, Gonzalez P, Aedo J, Estrada JM, Molina A, Valdés JA. Physiological and molecular responses to thermal stress in red cusk-eel (*Genypterus chilensis*) juveniles reveals atrophy and oxidative damage in skeletal muscle. *J Therm Biol.* 2020 Dec;94:102750.).

Overall, this is a very interesting and insightful manuscript and I recommend acceptance for publication.

Reviewer #3 (Remarks to the Author):

The authors present results regarding their work on identifying cold-induced FOXO1 nuclear transport and its role in cold tolerance of cell cultures, whole organisms – zebrafish; pre-diabetic mice exposed to 4°C for 14 days - and of murine and human pancreatic islets, with transplantation of murine pancreatic islets after 14 days of storage, re-establishing normoglycemia in diabetic mice for at least 2 consecutive days. Preservation of cell lines, organs and whole organisms is of major interest to the field, since the current limitation of storage is one of the key bottlenecks preventing usage of transplant organs. Preservation would improve logistics, worldwide matching of patients and allow for mixed chimerism protocols. Specifically to pancreatic islet preservation, diabetes is a major burden of disease for a large patient group where permanent treatment of this chronic disease is still lacking. Whilst Zhan et al. (1) showed successful vitrification of pancreatic islets, thereby potentially allowing infinite storage, mechanisms improving cold tolerance has the potential to increase survivability, and in vivo function of these vitrified pancreatic islets is yet to be shown. Furthermore, static cold storage requires less material and expertise than vitrification, allowing for easy implementation. Given the interdisciplinary nature of the described work, which requires bridging cryobiology, biomedical engineering and transplant surgery, it is also of interest to a wide audience that aligns well with Nature Communications in my opinion. The presented work shows translation of in vitro cold tolerance in multiple ex vivo and in vivo models with promising results. However, major concerns with the presented study consist of lacking control groups in all models and limited assessment of the recovery of the preserved cells/organ systems, specifically of pancreatic islets. Since metabolic health, particularly islet OCR, is predictive of islet function in vivo (2, 3) further assessments would be needed to assess overall success of the described

techniques. Furthermore, no histology or brightfield imaging is shown. Moreover, experimental set-up, number of animals or tissues used, and results are frequently showing omissions or are unclear.

Abstract

- Zebrafish not mentioned as a model system.

Title

- Consider changing “organ” to “tissue” as no experiments were performed on whole organs and only one tissue type, pancreatic islets was tested.

Introduction

- L66: please specify “few hours”.
- It might be worth mentioning/referencing Zhan et al. (1) storage of vitrified pancreatic islets.
- L80: Consider writing in passive voice.

Results

- It is unclear why the depicted cell lines were chosen.
- L99: It seems that cell death in H1 ESCs and iPSC-derived neurons after cold exposure was assessed after 2 hours, however it is known that damage following cold exposure can worsen or improve significantly after longer periods of recovery. Therefore reassessment at least 24h after rewarming would provide more accurate information on cell survival.
- L165: please correct ‘stie’
- L196: Is there a reason ‘Foxo1a’ is spelled differently here from the rest of the text (no capitals)? Same question for supplemental figure 7 and L56 in supplementary methods.
- L197: It is unclear how ‘significantly more’ abnormalities were quantified and how long after rewarming they were assessed. In the supplementary methods it is described assessment took place ‘after rewarming’ (how long after exactly?), but in supplemental figure 7 images are shown at 4°C. Furthermore, 72 hpf is said to be chosen as the experimental group due to consistently good cold tolerance, however results of 36-48 hpf are also shown, both displaying cold-induced pericardiac edema.
- Cold tolerance of pre-diabetic mice was investigated using two experimental groups: KPT-330 (inhibiting XPO1, thereby promoting FOXO1 nuclear accumulation) versus vehicle control group. However, positive control group(s) are missing. It would also be of interest if the cold tolerance can also be shown in healthy mice.
- L208: Given the toxicity of DMSO it would be of importance to mention the dosages used in the vehicle control group. In the supplementary methods this is only mentioned for the cultures, not for the in vivo experiments.
- L208: Please clarify that the mice are exposed to 4°C environment as this is not necessarily their body temperature during the exposure time.
- L210: It is unclear how lipid oxidation was measured from O₂-consumption and CO₂-production.
- L230: It is unclear why 14-day storage was chosen. There is no clear comparison shown between with and without protease inhibitors nor of shorter, but especially longer storage durations. This would be able to show a predicted viability for each storage durations and the limits of the techniques used. There are also no assessments shown at different stages of the preservation protocol (before, during, after storage). Furthermore, as mentioned earlier the viability assessments are limited and should be

repeated after extended recovery time (not merely right after rewarming). Moreover, no fresh control or sham transplantation was shown in vivo. The number of animals that have undergone transplantation is unclear. It is unclear whether other temperatures explored besides 4°C.

- L267: Please remove the ‘_’ before ‘Human’.

Supplementary methods

- General: the structure of the supplementary information does not follow the structure of the manuscript which makes it harder to find information.

- L161: ‘using’ instead of ‘by’ Prism.

Figures

- General:

o The method used for the depicted imaging/staining should be described in the figure text, this is not always clear.

o Displayed graphs are generally very small and hard to interpret with the current colours/legends/symbols. They would benefit from improvement and restructuring to make interpretation easier for the reader.

o In some figure descriptions there is no space between the figure number and ‘|’.

o Fresh controls are frequently omitted or a description of what the control group consists of.

o It is very frequently unclear what the n was for an analysis, either they are not reported or seem not to be in accordance with the n listed in the manuscript or figure text.

o Statistical analysis/significance is very frequently omitted from graphs, both in the manuscript and the supplemental material.

- Figure 6:

o C: mentions n=20 in the DMSO group in the graph legend, however in the text of D n=10 of which 5 died whilst the mortality graph C shows a mortality of about 75% meaning about 15 should be dead. It is unclear how these differences can be explained.

- Figure 7:

o Please note the type of imaging used.

o A: fresh control image would be useful for comparison.

o A: only significance between UW+DMSO and UW+KPT330 and between UW+KPT330 and HS+KPT330 is shown, but not between UW+DMSO and HS+KPT330, which seems unlikely optically.

o B: It is unclear why viability at 7 days is shown. Furthermore, insulin levels are shown 3h after recovery, however viability should also be assessed longer after storage than 3 hours to assess long term function.

o C: Statistical analysis of the fresh control group and UW 7d storage group is missing. It is unclear why 7d storage was chosen instead of a time matched control. It is unclear what K&P stands for.

- Figure 8:

o Unclear what the control conditions were.

o No 14-day storage images/data shown whilst L234 reads “Similar results were obtained from human donor pancreatic tissues and islets” whilst the difference between 2 and 14 days is significant.

o B: Line shown between UW24h and HS+K48h but no annotation.

- Supplemental figure 7:

o Please add descriptions of all abbreviations used.

- Supplemental figure 8

- o B: consider plotting means with standard deviation instead of one separate line per replicate.
- Supplemental figure 9
- o What does n=10 'trials' mean? Ten groups with each n=1 biological replicate or one group with n=10 biological replicates for each depicted group?
- o It seems there are fewer mice that were subjected to the HFD (n=15) and SD (n=10) than mice that entered the study as figure 6 shows n=16 in the KPT-330 group and n=20 in the DMSO group. It is unclear how this discrepancy can be explained.

References

1. Zhan, L., Rao, J.S., Sethia, N. et al. Pancreatic islet cryopreservation by vitrification achieves high viability, function, recovery and clinical scalability for transplantation. *Nat Med* 28, 798–808 (2022). <https://doi.org/10.1038/s41591-022-01718-1>
2. Papas, K. K. et al. Islet oxygen consumption rate (OCR) dose predicts insulin independence in clinical islet autotransplantation. *PLoS ONE* 10, e0134428 (2015).
3. Papas, K. K. et al. Human islet oxygen consumption rate and DNA measurements predict diabetes reversal in nude mice. *Am. J. Transplant.* 7, 707–713 (2007).

Reviewer #4 (Remarks to the Author):

The study by Zhang et al. investigates the role of the transcription factor FOXO1 in cold stress response and survival. They found that upon exposure to cold, FOXO1 translocates from the cytosol to the nucleus in human embryonic stem cells (ESCs). This process is mediated by the SUMO1 E3 ligase subunit RANBP2, importin-7 (IPO7) and exportin-1 (XPO1). FOXO1 regulates transcription of novel target genes during rewarming, affecting rhythmic processes, protein phosphorylation and RHO GTPase activities, as confirmed by ATAC-seq and integrated CUT&Tag and RNA-seq analyses.

In vivo experiments in zebrafish and obese mice, manipulation of FOXO1 localization by the XPO1 inhibitor KPT-330 increased cold tolerance in pre-diabetic obese mice. The study also found that targeting the XPO1-FOXO1 axis improved cold storage of pancreatic islets, which could help extend the shelf life of transplantable organs in clinical settings.

While the manuscript provides a comprehensive analysis of the role of FOXO1 in cold adaptation and its potential applications, investigating the contribution of different thermogenic tissues to cold protection in obese mice may provide valuable insights into the underlying mechanisms. Here are suggestions for experiments that the authors could perform:

(1) Evaluate the expression of known thermogenic markers (e.g., UCP1, PGC-1 α , PRDM16, SERCA1/2) in different thermogenic tissues, including brown adipose tissue (BAT), white adipose tissue (WAT), and skeletal muscle, to identify potential tissue-specific differences in response to cold exposure.

(2) Investigate the expression and localization of FOXO1 in these thermogenic tissues to determine if

there are tissue-specific differences in its regulation during cold stress.

(3) Investigate the potential role of other non-cell autonomous players (e.g., sympathetic nervous system, hormones) in altering the thermogenic response to cold stress by KPT-330.

(4) Evaluate metabolic flexibility and mitochondrial function in different thermogenic tissues of obese mice during cold exposure to identify potential alterations in energy metabolism that may contribute to their cold adaptation.

Finally, the authors should provide access to their NGS data GSE185152 to editors and reviewers.

Response to reviewers' comments

Firstly, we would like to thank our reviewers for your critical and constructive comments on our manuscript. Despite significant personnel changes in Ou's lab, we have made sure our new data are solid, and hope you find our efforts sufficient to address your concern and conclude the current story. Below are our point-to-point response to your comments:

Reviewer #1 (Remarks to the Author):

In this manuscript Zhang et al. aim to reveal the molecular mechanisms being relevant for cold-induced translocation of FOXO1. In my review, and given my field of expertise, I focus on the structural aspects of the manuscript. After having tested FOXO1 phosphorylation and acetylation in order to explain its nuclear retention, the authors aim to come up with alternative explanations. Overall, in its present form, the line of argumentation is difficult to follow and the statements are far-fetched in my opinion.

To me it seems that the bottom line of this part of the manuscript is that XPO1/CRM1 and IPO7 are important for re-localization of FOXO1. However, the role of SUMOylation or why the authors need SUMOylation at all to explain FOXO1 translocation is entirely unclear to me. Furthermore, it is unclear how the authors came up with the SUMO-interacting motifs for FOXO1 as the motif is an extremely short sequence and the manuscript lacks a convincing explanation for how this binding site was found and how it was determined. In line with this, the MD results and their interpretation seem far-fetched.

Response:

Your points are respectfully received. We could have presented our data in a better order for readers. This story is largely based on experimental models, and Dr. Qingfen Yu is the only expert among us in MD modeling. After careful discussion with Dr. Yu and Dr. Wei Li's team at the National Institutes of Health (U.S.), we have rearranged **Fig. 4** to present experimental results first, then the PCA analysis on the FOXO1 N-terminus (containing a predicted SIM)/SUMO1 vs. FOXO3 (a transcription factor closely related to FOXO1) N-terminus (no predicted SIM)/SUMO1 at 37°C or 4°C. The PCA analysis supports our experimental data that FOXO1 SIM and SUMO1 have a more stable conformation at 4°C, whilst FOXO3 N-terminus/SUMO1 have much weaker interactions at 37°C or 4°C (please see **Fig. 4f** and **Supplementary Fig. 6**). Then in **Fig. 5**, we show that in the cell lines we used, endogenous FOXO3 proteins are largely localized in the cell nucleus independent of temperature, but once we overexpressed mutant FOXO3 proteins that carry the N-terminal

FOXO1 SIM, these mutant proteins can distribute in the cytosol at 37°C, and largely accumulate in the nucleus at 4°C (**Fig. 5c**).

The role of SUMOylation or why the authors need SUMOylation; how the authors came up with the SUMO-interacting motifs for FOXO1; explanation for how this binding site was found and how it was determined: there are 2 papers introducing online tools for the prediction of SIM and SUMOylation sites^{1,2}, and these tools both predict a N-terminal FOXO1 SIM, and SUMOylation sites on IPO7 and XPO1 (please see **L147-148** and **Fig. S4a**). These led us to do a lot of screening (not shown in this manuscript). Firstly, no strong evidence of direct SUMOylation on FOXO1 was found. Instead, we have accumulated data showing that the nuclear transport system RANBP2 (but not RANGAP1/UBC9), XPO1 and IPO7 are responsible for temperature-driven FOXO1 transport and temperature-sensitive differential interactions with SUMO1 but not SUMO2/3 (mainly in **Fig. 3** and **Fig.S4c-h**); our previous layout failed to bring to your attention (data are now in **Fig. 3g** and **Fig. 4c**) that such features are dependent on the predicted SUMOylation sites of IPO7 and XPO1: when mutant IPO7(K517R) or XPO1(K752R,K957R) with disrupted SUMOylation site(s) were overexpressed, the interaction between these mutant proteins and FOXO1 was much weakened. Additionally, **Fig. 4** mainly demonstrated the importance of FOXO1 SIM for the interaction of XPO1/FOXO1, and as stated above (**Fig. 5**), mutant FOXO3 proteins containing a N-terminal SIM demonstrated temperature-sensitive cellular distribution. We hope these data presented in the current layout are sufficient to convince you that SUMOylation sites on XPO1 and IPO7, and FOXO1 SIM are vital to the interaction between the transporter proteins and the cargo (FOXO1). Corresponding text has also been highlighted for your reading convenience.

Assuming XPO1/CRM1 and IPO7 as the key factors, and given that the NLS/NES of FOXO1 has already been identified, I'm wondering why the authors didn't mutate the FOXO1 NLS/NES and why they didn't use XPO1/CRM1 and IPO7 inhibitors to validate the interaction. For FOXO1 it has already been reported that phosphorylation of several residues by PKB/AKT, CK1 and DYRK can lead to nuclear exclusion. Not all of these sites have been evaluated by the authors. Seeing that that the DYRK site seems to become dephosphorylated after 4h at 4°C (SFig. 3A), I suggest that, next to the inhibitors, the authors test relocalization of P-mimicking (Asp or Glu) and P-deficient (Ala) FOXO1 mutants.

Response:

We sincerely thank your advice on the role of NLS/NES – it is indeed very helpful to determine how important SIM is in the context of our story. As shown in **Fig. 4d and e**,

overexpressed mutant FOXO1 proteins lacking both NLS and NES (based on PMID: 10377430) could not be transported into the nucleus at 4°C; however, FOXO1-minus NLS/NES proteins still could interact with XPO1 or IPO7 in patterns similar to those between WT FOXO1 proteins and XPO1 or IPO7. These new results are in clear conformation with our previous data and our model (**Fig. 5e**). To sum up, the actual FOXO1 protein nuclear import is still determined by the NLS, but the newly reported FOXO1 SIM here serves as a nuclear export signal at 37° C and a ‘temperature-sensitive’ code, altering FOXO1 interaction with SUMOylated XPO1 or IPO7 proteins at different temperatures.

Indeed, FOXO1 phosphorylation is a known mechanism for FOXO1 nuclear export. Hence, if one sees seemingly increased levels of p-FOXO1 at 4°C, the expectation should be more cytoplasmic distribution of FOXO1, whilst FOXO1 nuclear accumulation was seen in hibernator tissues/organs and human ESCs at 4°C at the same time of increased p-FOXO1 levels. Now, as shown in **Fig. S3b and c**, immunofluorescence of cellular endogenous p-FOXO1(S249), p-FOXO1(S256) and p-FOXO1(S319), or overexpressed FOXO1(T24D) and FOXO1(S329E) revealed predominantly nuclear signals at 4°C, suggesting such FOXO1 single-site phosphorylation is not sufficient to block cold-induced FOXO1 nuclear import. Please note that the FOXO1 SIM-RANBP2/XPO1/IPO7 mechanism reported in this manuscript is not in a mutually exclusive relationship with FOXO1 phosphorylation – a statement has been added (L280-285; Discussion). We hope to follow up in this direction when we have the resources, to evaluate which phosphorylation (and/or acetylation – see **Fig.S3g**) sites could also affect the efficiency of temperature-driven FOXO1 transport.

As for XPO1 inhibitor, we used KPT-330 in the rest of the manuscript, and we could not find an IPO7-specific inhibitor. We reckon extra experiment with KPT-330 in considering FOXO1-XPO1 interaction cannot serve the supposed purpose because the inhibitor reduced XPO1 functions at protein levels (please see **Fig. 6a** as an example). Again, please see **Fig. 4c**, demonstrating the importance of FOXO1 SIM and XPO1 SUMOylation sites (K752 and K957), and **Fig. 3g**, demonstrating the importance of IPO7 SUMOylation site (K514) in the interaction between cargo (FOXO1) and transporter proteins (XPO1/IPO7).

We hope our data and explanation above are sufficient to convince you that our model is reliable and novel to be accepted by the journal.

1. Zhao, Q. *et al.* GPS-SUMO: a tool for the prediction of sumoylation sites and SUMO-interaction motifs. *Nucleic Acids Res* **42**, W325–330 (2014).
2. Beauclair, G., Bridier-Nahmias, A., Zagury, J.F., Saib, A. &

Zamborlini, A. JASSA: a comprehensive tool for prediction of SUMOylation sites and SIMs. *Bioinformatics* **31**, 3483–3491 (2015).

Reviewer #2 (Remarks to the Author):

The manuscript clearly revealed the involvement of FOXO1 in regulating cold stress in human and mouse cells. The authors reported FOXO1 translocation from cytoplasm to nucleus with SUMO-modification to be the key to relay the cold signal and elicit the cold responses. A nonconventional, temperature-sensitive FOXO1 transport mechanism involving the nuclear pore complex protein RANBP2, Importin-7 and Exportin-1, and a SUMO-interacting motif on FOXO1 was found to control nuclear entry of FOXO1. More importantly, the authors found the Exportin-1 inhibitor KPT-330 greatly enhanced cold tolerance prolonged the shelf-life of human and mouse pancreases and islets. These findings are original and extending current understanding on mechanisms of cellular cold tolerance. The results have potential to promote cold therapy and cold preservation of human tissues as the authors have demonstrated in the paper.

The work is well executed and evidence is adequate to support the authors' claims. The methodology used are in the cutting edge of the field.

I do have a small concern that the authors might be able to clarify further:

1. Through Cut&Tag and RNA-seq integration analysis, the authors found that the Foxo1 binding motif was present in 20% of genes whose expression was significantly changed under cold treatment. However, I have a concern about the role of Foxo1 as the major contributor. Is it possible that other Foxo family members or transcription factors also participate in the regulation of the remaining 80% of genes? In this manuscript, the authors wrote: "Also, FOXO1 proteins were enriched in TLGS cell nuclei of multiple tissues and organs during hibernation or following storage at 4°C, while FOXO3 did not show such dynamics, suggesting a specific role of FOXO1 in cold adaptation during hibernation (Supplementary Fig. 7e-g)." No dynamic pattern as FOXO1 does not preclude the role of FOXO3 in cold response, as some literature indicated such involvement. Hu P, Liu M, Liu Y, Wang J, Zhang D, Niu H, Jiang S, Wang J, Zhang D, Han B, Xu Q, Chen L. Transcriptome comparison reveals a genetic network regulating the lower temperature limit in fish. *Sci Rep.* 2016 Jun 30;6:28952. doi: 10.1038/srep28952. AND other studies reveal the involvement of FOXOs in thermal responses. (Dettleff P, Zuloaga R, Fuentes M, Gonzalez P, Aedo J, Estrada JM, Molina A, Valdés JA. Physiological and molecular responses to thermal stress in red cusk-eel (*Gnypterus chilensis*) juveniles reveals atrophy and oxidative damage in skeletal muscle. *J Therm Biol.* 2020 Dec;94:102750.).

Overall, this is a very interesting and insightful manuscript and I recommend acceptance for publication.

Response:

Although our response to you is short, our gratitude to your support and positive view on our work is immense! Following your suggestion, we have revised our Discussion section. Please see highlighted parts in Discussion.

Reviewer #3 (Remarks to the Author):

The authors present results regarding their work on identifying cold-induced FOXO1 nuclear transport and its role in cold tolerance of cell cultures, whole organisms – zebrafish; pre-diabetic mice exposed to 4°C for 14 days - and of murine and human pancreatic islets, with transplantation of murine pancreatic islets after 14 days of storage, re-establishing normoglycemia in diabetic mice for at least 2 consecutive days. Preservation of cell lines, organs and whole organisms is of major interest to the field, since the current limitation of storage is one of the key bottlenecks preventing usage of transplant organs. Preservation would improve logistics, worldwide matching of patients and allow for mixed chimerism protocols. Specifically to pancreatic islet preservation, diabetes is a major burden of disease for a large patient group where permanent treatment of this chronic disease is still lacking. Whilst Zhan et al. (1) showed successful vitrification of pancreatic islets, thereby potentially allowing infinite storage, mechanisms improving cold tolerance has the potential to increase survivability, and in vivo function of these vitrified pancreatic islets is yet to be shown. Furthermore, static cold storage requires less material and expertise than vitrification, allowing for easy implementation. Given the interdisciplinary nature of the described work, which requires bridging cryobiology, biomedical engineering and transplant surgery, it is also of interest to a wide audience that aligns well with Nature Communications in my opinion.

The presented work shows translation of in vitro cold tolerance in multiple ex vivo and in vivo models with promising results. However, major concerns with the presented study consist of lacking control groups in all models and limited assessment of the recovery of the preserved cells/organ systems, specifically of pancreatic islets. Since metabolic health, particularly islet OCR, is predictive of islet function in vivo (2, 3) further assessments would be needed to assess overall success of the described techniques. Furthermore, no histology or brightfield imaging is shown. Moreover, experimental set-up, number of animals or tissues used, and results are frequently showing omissions or are unclear.

Response:

We truly admired your critical, proofread-level review. Here in summary, we have performed islet OCR as you suggested; the main reason for the seemingly ‘lacking control groups’ or ‘limited assessment of the recovery’ was because islets cold-stored in UW solution for longer than 7 days would tend to auto-lyse after rewarming and culturing at 37°C for longer than 2 h. Similarly, H1 ESCs rewarmed for longer than 2 h will often lead to detachment of colonies. Please see our specific responses below, and hope you would find our new data, revision and explanation satisfactory.

Abstract

- Zebrafish not mentioned as a model system.

Zebrafish is mentioned in the abstract now, changes highlighted.

Title

- Consider changing “organ” to “tissue” as no experiments were performed on whole organs and only one tissue type, pancreatic islets was tested.

Title is changed per your suggestion.

Introduction

- L66: please specify “few hours”.

‘a few hours’ has been changed to ‘about 10 hours’.

- It might be worth mentioning/referencing Zhan et al. (1) storage of vitrified pancreatic islets.

Please see updated and highlighted text on **L76-80**.

- L80: Consider writing in passive voice.

The sentence has been changed to: ‘Consistently, good tolerance to cold exposure has been shown in human embryonic stem cells (hESCs)’.

Results

- It is unclear why the depicted cell lines were chosen.

Firstly, hESCs tolerate 4°C exposure relatively well. Our results show that hESCs well retain the temperature-driven FOXO1 transport phenotype (**Fig.1d** and extra data in figure presented below). As shown in Fig. S1c and d, human infant iPSC-derived neurons also manifested such a phenotype and good cold tolerance, but it quickly deteriorated in cells from higher passages; adult iPSC-derived neurons no longer responded to cold exposure with the initiation of FOXO1 nuclear import and had poor cold survival. We also tested many cancer (not shown) or immortalized cell lines (such as the ARPE-19 cell line), and confirmed that many of these immortalized cell lines (Not ARPE-19) have constitutive, high-level nuclear FOXO1 and good cold tolerance, but we did not want to count on these cell lines for our mechanistic works. Hence, hESCs were our preferred choice. Our attempts to use CRISPR to mutate FOXO1 SIM in H1 ESCs failed, probably because normal FOXO1 functions are essential to pluripotent stem cells (please see **L175-179**). Instead, we had to rely on ARPE-19, a relatively normal human cell line to conduct our mechanistic works on FOXO1 SIM. It had been indeed rather challenging for us to settle on suitable human cell lines/types, and it is hard to explain this in the story. Hope our background history on choosing the cell models is clear

to you now.

- L99: It seems that cell death in H1 ESCs and iPSC-derived neurons after cold exposure was assessed after 2 hours, however it is known that damage following cold exposure can worsen or improve significantly after longer periods of recovery. Therefore reassessment at least 24h after rewarming would provide more accurate information on cell survival.

We agree with you that cold-induced injury, or recovery from cold after longer periods should be evaluated, but perhaps not as a major aspect in the context of this story due to these reasons: 1) in temperature-induced FOXO1 transport (for example, see **Fig. 1f, g, Fig. S1c, and Fig. S7e**), the majority of FOXO1 proteins would have been exported out of the cell nucleus within 2 hours; phenotypes occurred after 24 h could be regulated by FOXO1, or could be secondary and accumulated effects regulated by other factors; 2) as stated above and shown below: for human iPSC-derived neurons, at 8-h after rewarming, no exacerbated cell death (Propidium Iodide staining; red) or neurite (TUBB3 staining; magenta) regeneration could be found (this is the nature of neurons though), hence it was unclear how we should evaluate these neurons; as for hESCs, they recovered well with no exacerbated cell death, but cell attachment property indeed worsened, so live cell mass routinely detached and affected our observations; overall, human stem cell-based models were still too delicate to support longer term recovery surveys; 3) in this story, we want to focus on the temperature-driven FOXO1 transport and cold survival phenotype, the FOXO1 SIM as a regulatory element that differentially interact with SUMOylated IPO7 or XPO1 at different temperatures, and using XPO1 inhibitor KPT-330 could improve cold survival in obese mid-age mice. The zebrafish morpholino-knock down of *foxo1* (**Fig. S7d**) and survival after cold exposure experiments, and the islet storage and transplantation experiments in this story can serve as *in vivo* evaluation of long-term recovery from extreme cold stress.

To sum up, we do not think cultured cell models are ideal for longer term evaluation on recovery from extensive cold stress, and we already provide clear *in vivo* data to reflect long-term recovery. We hope you would accept our proposal to leave this issue to a separate story.

Confocal micrographs showing, a) iPSC-neurons from an infant donor, and b) H1 ESCs stained with antibodies against FOXO1 (green), Propidium Iodide (red; PI), neuron-specific TUBB3 (magenta) or DAPI (blue), at annotated conditions. Note that in these 2 cell types, after 8-h of recovery, no significantly elevated cell death was seen. However, it was difficult to evaluate their cellular physiology as the iPSC-neurons had difficulty to regenerate neurites (common to neurons), and live hESCs tended to detach from the well.

- L165: please correct 'stie'

Sorry for the typo.

- L196: Is there a reason 'Foxo1a' is spelled differently here from the rest of the text (no capitals)? Same question for supplemental figure 7 and L56 in supplementary methods.

According to <http://zfin.org/ZDB-GENE-061013-59#summary>, it seems we should not even capitalize 'F' for zebrafish gene *foxo1a*. As for the protein nomenclature, we also should write as Foxo1a (<https://zfin.atlassian.net/wiki/spaces/general/pages/1818394635/ZFIN+Zebrafish+Nomenclature+Conventions#ZFINZebrafishNomenclatureConventions-2>). Changes have been made.

- L197: It is unclear how 'significantly more' abnormalities were quantified and how long after rewarming they were assessed. In the supplementary methods it is described assessment took place 'after rewarming' (how long after exactly?), but in supplementary figure 7 images are shown at 4°C. Furthermore, 72 hpf is said to be chosen as the experimental group due to consistently good cold tolerance, however results of 36-48 hpf are also shown, both

displaying cold-induced pericardiac edema.

Sorry for the ambiguity. In our hands, apparent cold-induced abnormalities in zebrafish larvae would have occurred by the end of 4°C incubation, and we will do the microscopic examination and imaging during the 2-h rewarming (**Fig.S7d**). Now, in Supplementary methods, this description is provided as follows: ‘followed by rewarming to 28°C for 2 h and microscopic imaging on larva survival and morphology. Abnormal or dead larvae were removed, and the rest of the larvae were allowed to grow to adulthood with weekly inspections for any apparent developmental defects.’

As for the rest of the questions you mentioned here, firstly, please see **Fig. S7d**: in our hands, at 36-47 hpf, higher percentage of zebrafish larvae manifested cold-induced pericardiac edema (some are mild and reversible) and abnormal irreversible body curvatures. We decided to show in **Fig. S7a** a 72-hpf zebrafish larva with mild cold-induced pericardiac edema because we wanted to show that 4°C could still cause mild phenotype at this stage, but rarely. Last but not least, ‘significantly more’ (now **L219**): the sentence has been changed to ‘cold-exposed larvae more frequently manifested pericardiac edema and severe and irreversible abnormal body curvatures’.

- Cold tolerance of pre-diabetic mice was investigated using two experimental groups: KPT-330 (inhibiting XPO1, thereby promoting FOXO1 nuclear accumulation) versus vehicle control group. However, positive control group(s) are missing. It would also be of interest if the cold tolerance can also be shown in healthy mice.

Your comment is well accepted. However, we argue that 1) cold tolerance in healthy, younger mice has been done and reported by many other labs; 2) the physiology of healthy, young adult mice is apparently different from that of the mid-age, obese and pre-diabetic mice, so it is unclear to what extent these healthy mice can be ‘positive controls’ in the context of KPT-330 treatment; 3) nonetheless, the Ou lab intends to follow the suggestions from you and reviewer #4, to systematically compare cold tolerance in healthy mice with that in health-compromised mice. Thus, instead of adding more ‘control’ mouse experiments, this would be better a fit as the foundation for another work.

- L208: Given the toxicity of DMSO it would be of importance to mention the dosages used in the vehicle control group. In the supplementary methods this is only mentioned for the cultures, not for the in vivo experiments.

Text in the Supplementary methods has been updated as: ‘For KPT-330 treatment, mice were injected i.p. with KPT-330 (Selleck, 5 mg/kg) 30 min prior to acute cold stimulation; for vehicle control, i.p. injection with equal amount of DMSO (1 ml/kg) was performed’.

- L208: Please clarify that the mice are exposed to 4°C environment as this is not necessarily their body temperature during the exposure time.

The sentence (now **L231**) has been updated: ‘one injection of 5 mg/kg KPT-330 aided obese middle-aged mice to adapt and survive for 10 days of cold exposure at 4°C,’. We also added an ‘Ambient temperature’ label on **Fig. 6c**.

- L210: It is unclear how lipid oxidation was measured from O₂-consumption and CO₂-production.

Sorry about this. The sentence you mentioned was from a previous version of the manuscript and we forgot to correct it. Now the statement is (now **L233**): ‘When cold exposure started, KPT-330 treatment appeared to facilitate heat production, O₂ consumption and CO₂ production;’.

- L230: It is unclear why 14-day storage was chosen. There is no clear comparison shown between with and without protease inhibitors nor of shorter, but especially longer storage durations. This would be able to show a predicted viability for each storage durations and the limits of the techniques used. There are also no assessments shown at different stages of the preservation protocol (before, during, after storage). Furthermore, as mentioned earlier the viability assessments are limited and should be repeated after extended recovery time (not merely right after rewarming). Moreover, no fresh control or sham transplantation was shown *in vivo*. The number of animals that have undergone transplantation is unclear. It is unclear whether other temperatures explored besides 4°C.

1) In the literature (for example, PMID: 20061918 and 8278997), experimental static cold storage of mouse islets in UW solution at 4°C typically lasts for 2-3 days. In our preliminary experiments, mouse islets stored at 4°C in UW solution for 5-7 days could maintain their morphology after rewarming (islet disintegration during cold storage and rewarming occurred but was mild), but they barely recovered islet functions; for islets stored for 10 or more days, they usually would disintegrate soon after rewarming. Our advance reported here could steadily protect the islets for 2 weeks, hence for the sake of completing and publishing this story, we settled for 14-day storage. In our opinion, this primarily is not a technical paper, and there could be room in some other details that can push the cold storage time further in a follow-up work. We hope you would agree with us in this regard.

2) The protective effects of protease inhibitors has been reported by us previously (PMID: 29576452). Even for islets stored in HS + KPT-330, > 7-day storage usually had exhausted the nutrient storage and the proteome of the islet cells, so IF staining of certain targets could yield much weaker signals, and *in vitro* rewarming and functional tests imposed stress on this

extendedly stored islets. Visual difference of the morphology of mouse and human islets stored for 14 days with or without protease inhibitors was obvious (please see **Fig. S9f** and **S10d**).

3) We followed your suggestion to perform islet OCR measurements to provide extra assessments on the islet quality (please see **Fig. 7c**). Please note that mammalian cells and tissues exposed to 4°C still have life activities, especially the degradation machinery. From the normalized OCR results, it is apparent that islets stored in UW solution for 7 days could barely recover (very low OCR levels), and those stored in UW solution for 14 days could not recover; however, islets stored with our method also had low OCR levels compared to that of fresh controls, suggesting their recovery needs longer time like you mentioned, but in vitro culture was not a good option. Instead, our *in vivo* transplantation experiments (**Fig. 7e** and **f**) provide strong evidence that islets stored with our method could recover and function for a long time in mice, but those stored in UW solution could not.

4) For mouse pancreases (**Fig. S9a**) and human pancreatic samples (**Fig. 8a and b**), we showed the results from cold-stored 48 h, or 24 and 48 h, respectively. Assessment results for mouse isolated islets stored for 2 days (48 h; **Fig. 7a, S9b** and **c**), 7 days (**Fig. 7b** and **S9d**), and 14 days (**Fig. 7c** and **S9e**), and human isolated islets stored for 2 days (**Fig. S10a** and **b**) and 14 days (**Fig. 8c, Fig. S10c** and **d**).

5) New transplantation data (**Fig. 7e**) and text have been updated and highlighted. No other temperature was tested. Nonetheless, the successful publication of this story would lay the foundation for us to team up with our clinician colleagues and work more specifically for the translational research on human islet cold storage and transplantation.

- L267: Please remove the ‘ ’ before ‘Human’.

Again, we deeply appreciate your help in correcting our typos and mistakes!

Supplementary methods

- General: the structure of the supplementary information does not follow the structure of the manuscript which makes it harder to find information.

We have tried to restructure the order of Supplementary methods. New data per suggestions from you and other reviewers have been provided, figures updated. This work contains broad spectrum of data, from bioinformatics and cell biology to mouse islet transplantation. We would be happy to follow specific instructions from our reviewers and the editors if further improvement is needed.

- L161: 'using' instead of 'by' Prism.

Done!

Figures

- General:

o The method used for the depicted imaging/staining should be described in the figure text, this is not always clear.

Done! Changes highlighted in legend.

o Displayed graphs are generally very small and hard to interpret with the current colours/legends/symbols. They would benefit from improvement and restructuring to make interpretation easier for the reader.

Some changes have been made, but space limit is indeed an issue. Could we please leave these to the editorial office and us? I (Jingxing Ou) am not a professional in this, whilst the first 5 authors all have started new chapters of their life, so it would be easier if we can address this at the later stage.

o In some figure descriptions there is no space between the figure number and ' '.

Done, thank you!

o Fresh controls are frequently omitted or a description of what the control group consists of.

Because no specific is mentioned here, we speculate you were referring to mostly the mouse cold exposure and islet results. We should have already covered these aspects above.

o It is very frequently unclear what the n was for an analysis, either they are not reported or seem not to be in accordance with the n listed in the manuscript or figure text.

Changes have been made and highlighted in the legend. The 'discrepancy' of n values in Fig. 6 is because we conducted multiple survival experiments, but not all mice were subjected to monitoring by the metabolic cage system. Unnecessary annotation has been deleted. The n for DMSO group in mouse body mass was indeed a typo. It should be 5 (the 5 survivors from the total of 20 mice in the DMSO group) and has been changed.

o Statistical analysis/significance is very frequently omitted from graphs, both in the manuscript and the supplemental material.

We only showed the statistical significance from comparisons we thought were essential, otherwise the figure panels would be too busy to read.

- Figure 6:

o C: mentions n=20 in the DMSO group in the graph legend, however in the text of D n=10 of which 5 died whilst the mortality graph C shows a mortality of about 75% meaning about 15 should be dead. It is unclear how these differences can be explained.

Our previous legend description was indeed confusing. As explained above, the survival data were the add-up total from our multiple experiments. The results from **Fig. 6d** were from 'a representative experiment' as we stated, and the n for DMSO was clarified as n = 5 (by the end of measurement, 5 mice were still alive, and only their data were used). The description for **Fig. 6e** was updated, as the n numbers are the add-up total of mice subjected to metabolic cage monitoring and survived till the end of experiment. That is to say, we had 7 mice in the KPT-330 treatment group that were **not** put into the metabolic cage system due to space limitation. Likewise, we had very limited access to the instrument for measuring mouse body mass composition (**Fig. 6f**), and we did what we could.

- Figure 7:

o Please note the type of imaging used.

Done!

o A: fresh control image would be useful for comparison.

Considering the space limitation, we show the staining results of fresh islets in **Fig. S9b, d and f**. We also show the fresh control results of human islets in **Fig. S10a, b and d**.

o A: only significance between UW+DMSO and UW+KPT330 and between UW+KPT330 and HS+KPT330 is shown, but not between UW+DMSO and HS+KPT330, which seems unlikely optically.

As explained above, we only showed the comparison we deemed essential for our readers, or the panels can easily become overwhelming. If there are specific places that you or the editors want us to provide statistical significance, we will be happy to do so at a later stage.

o B: It is unclear why viability at 7 days is shown. Furthermore, insulin levels are shown 3h after recovery, however viability should also be assessed longer after storage than 3 hours to assess long term function.

As explained above, mouse islets stored in UW for 7 or more days could not recover, whilst longer *in vitro* 'normal incubation' condition is stressful for islets after prolonged cold storage (the *in vitro* results would seemingly be 'ugly'). This and our consideration to restrict our use on animal resources (each experimental condition could easily consume 6-10 mice)

made us decide not to perform islet functional assessments at later recovery time points.

o C: Statistical analysis of the fresh control group and UW 7d storage group is missing. It is unclear why 7d storage was chosen instead of a time matched control. It is unclear what K&P stands for.

We have provided statistical results. UW 7-d was chosen, because islets from UW 14-d often had severe auto-lysis during cold storage and after rewarming, so it was impractical to do so. K&P was explained in the legend as KPT-330 and protease inhibitors.

- Figure 8:

o Unclear what the control conditions were.

The annotation has been updated as 'Fresh ctrl', meaning fresh pancreatic tissues or islets we freshly isolated from these tissues.

o No 14-day storage images/data shown whilst L234 reads "Similar results were obtained from human donor pancreatic tissues and islets" whilst the difference between 2 and 14 days is significant.

We do have 14-day islet storage results – please see **Fig. 8c, S10c and d**. To avoid ambiguity, the sentence has been changed to: 'Similar results were obtained from human donor pancreatic tissues (up to 2 days) and islets (up to 14 days) isolated from them (Fig. 8a-c and Supplementary Fig. 10).' Please note that due to ethical regulations and limits to tissue availability, we can not perform extensive assessments and staining experiments at the moment.

o B: Line shown between UW24h and HS+K48h but no annotation.

Lines have been updated. The purpose of comparing UW24h and HS+K48h is to show that HS+K48h had better tissue protection than that of UW24h.

- Supplemental figure 7:

o Please add descriptions of all abbreviations used.

We only spotted the description for TLGS was missing (now added).

- Supplemental figure 8

o B: consider plotting means with standard deviation instead of one separate line per replicate.

One line for each animal was because in this way people can see (after zooming in) when the 7 mice in the DMSO group were deceased during the experiment.

- Supplemental figure 9

o What does n=10 ‘trials’ mean? Ten groups with each n=1 biological replicate or one group with n=10 biological replicates for each depicted group?

Legend has been updated.

o It seems there are fewer mice that were subjected to the HFD (n=15) and SD (n=10) than mice that entered the study as figure 6 shows n=16 in the KPT-330 group and n=20 in the DMSO group. It is unclear how this discrepancy can be explained.

We purchased batches of obese mouse model (12-14 month) from our animal provider with the specified conditions. They delivered the animals to us with their certificate. We then selected 15 of them for verification. Animal provider information was added in Methods.

References

1. Zhan, L., Rao, J.S., Sethia, N. et al. Pancreatic islet cryopreservation by vitrification achieves high viability, function, recovery and clinical scalability for transplantation. *Nat Med* 28, 798–808 (2022). <https://doi.org/10.1038/s41591-022-01718-1>
2. Papas, K. K. et al. Islet oxygen consumption rate (OCR) dose predicts insulin independence in clinical islet autotransplantation. *PLoS ONE* 10, e0134428 (2015).
3. Papas, K. K. et al. Human islet oxygen consumption rate and DNA measurements predict diabetes reversal in nude mice. *Am. J. Transplant.* 7, 707–713 (2007).

Reviewer #4 (Remarks to the Author):

The study by Zhang et al. investigates the role of the transcription factor FOXO1 in cold stress response and survival. They found that upon exposure to cold, FOXO1 translocates from the cytosol to the nucleus in human embryonic stem cells (ESCs). This process is mediated by the SUMO1 E3 ligase subunit RANBP2, importin-7 (IPO7) and exportin-1 (XPO1). FOXO1 regulates transcription of novel target genes during rewarming, affecting rhythmic processes, protein phosphorylation and RHO GTPase activities, as confirmed by ATAC-seq and integrated CUT&Tag and RNA-seq analyses.

In vivo experiments in zebrafish and obese mice, manipulation of FOXO1 localization by the XPO1 inhibitor KPT-330 increased cold tolerance in pre-diabetic obese mice. The study also found that targeting the XPO1-FOXO1 axis improved cold storage of pancreatic islets, which could help extend the shelf life of transplantable organs in clinical settings.

While the manuscript provides a comprehensive analysis of the role of FOXO1 in cold adaptation and its potential applications, investigating the contribution of different thermogenic tissues to cold protection in obese mice may provide valuable insights into the underlying mechanisms. Here are suggestions for experiments that the authors could perform:

Response:

Before our point-to-point response to your comments, we would first like to thank you for your suggestions on further experiments we can perform. What you suggested us to do were actually what we wanted to follow up after the conclusion of this story. The Ou lab was derived from the Wei Li lab at the NIH (retina; hibernation) and has been hoping to bridge hibernation research with translational medicine (in this story, FOXO1 and cold adaptation, islet preservation and potential hypothermic treatments to certain pre-diabetic conditions), and the rest of the authors are students/postdocs/clinicians/a computational biologist. We want to conduct an in-depth, comprehensive study following the direction of your suggestion, which would need a lot of time, resources and new expertise in the BAT/WAT/muscle and thermogenesis field. Hence, in order to respect and respond to your comments and keep our current story on track, we mainly conducted new cold-exposure experiments in obese prediabetic mice, then collected their BAT, WAT, skeletal muscle and liver samples, and conducted transcriptomic analyses, hoping to provide new insights you and readers of our manuscript may find valuable. Please note that in the new **Supplementary Data 3**, *Xpo1* was differentially expressed in the KPT330-treated vs. DMSO-treated comparisons in all 4 types of tissues we surveyed, which is a strong indication that KPT-330 had taken effects in these tissues. Our qPCR validation (**Fig.S8c**) also proves that these mice were indeed exposed to cold. Interestingly, pathway enrichment analysis (**Fig. 6h** and **S8g-j**) indicates that other than

complex metabolic changes, immune responses were triggered particularly in eWAT and skeletal muscles of the cold-exposed prediabetic mice, whilst KPT-330 treatment modified heterochromatin formation and transcription factor activity in eWAT, and stress and immune responses in all 4 types of tissues. FOXO1 is a key transcription factor in metabolism and stress regulation, and important to the development/differentiation of immune cells. Like we suggest here, more specific mouse models may be required in a new project to further elucidate these profound phenomena.

(1) Evaluate the expression of known thermogenic markers (e.g., UCP1, PGC-1 α , PRDM16, SERCA1/2) in different thermogenic tissues, including brown adipose tissue (BAT), white adipose tissue (WAT), and skeletal muscle, to identify potential tissue-specific differences in response to cold exposure.

Accordingly ^{1,2}, qPCR of *Ucp1*, *Pparg1a*, *Cidea*, *ELovl3*, *Cox8b* and *Prdm16* was performed (**Fig.S8c**). The expression of other genes potentially related to thermogenesis can be found in our new RNA-seq data. Consistent with the pathway enrichment analysis mentioned above, KPT-330 treatment only modestly further enhanced the expression of *Cidea* in eWAT and muscle, and *Pparg1a* in muscle. Thus, KPT-330 may not directly improve thermogenesis in these health-compromised mice, but in regulating the immune and stress responses. Please see highlighted revised text in Results and Discussion sections.

(2) Investigate the expression and localization of FOXO1 in these thermogenic tissues to determine if there are tissue-specific differences in its regulation during cold stress.

We have tried many times with different protocols and antibodies. Unfortunately, FOXO1 IF or IHC staining in BAT and eWAT has been unsuccessful in our hands. FOXO1 IF staining results in skeletal muscle are now presented in **Fig. 6b**, together with the FOXO1 IF staining in heart and liver sections. Together with our new multiple tissue RNA-seq data, it is apparent that KPT-330 can effectively promote FOXO1 nuclear transport in the pancreas, heart, liver and skeletal muscle (**Fig. 6a and b**).

(3) Investigate the potential role of other non-cell autonomous players (e.g., sympathetic nervous system, hormones) in altering the thermogenic response to cold stress by KPT-330.

As stated above, we currently do not have the expertise, resources (for example, more specific mouse models) and time to provide detailed analysis on the non-cell autonomous players in this context. Nonetheless, as shown in **fig. S8g-j** (DEGs in these enriched pathways can be found in **Supplementary Data 3**), in these obese, prediabetic, mid-age mice, acute cold exposure up-regulates genes related to peptide hormone pathway in BAT and cytokine-

related signaling in eWAT; in comparison (**Fig.6h**), KPT-330 up-regulates genes related to peptide hormone pathway in eWAT, genes in the cytokine-related pathways in skeletal muscles and liver. Although no pathways related to sympathetic nervous system were found among the top enriched pathways, we intend to follow up in these aspects in the future.

(4) Evaluate metabolic flexibility and mitochondrial function in different thermogenic tissues be found of obese mice during cold exposure to identify potential alterations in energy metabolism that may contribute to their cold adaptation.

As shown in **fig. S8g-j**, in the mice we tested, acute cold exposure up-regulates genes in the lipid metabolism pathways in BAT, and various metabolic pathways in skeletal muscle, and metabolism of RNA and amino acids in the liver; in comparison (**Fig.6h**), other than up-regulated proteolysis in BAT or protein ubiquitination/acetylation in eWAT, KPT-330 treatment appeared to mainly affect stress and immune responses. Nonetheless, metabolic adaptation and alteration in mitochondrial metabolic activities in the immune system would be a very interesting topic to pursue in the future.

Finally, the authors should provide access to their NGS data GSE185152 to editors and reviewers.

Token to access GSE185152 is: [svjkoachrglnej](https://www.ncbi.nlm.nih.gov/geo/query/acc.cgi?acc=GSE185152). Please note that it also includes the new RNA-seq data from cold-exposed obese mice, which should be finished uploading to the project by the time you read this.

1. Qiu, Y. *et al.* Eosinophils and type 2 cytokine signaling in macrophages orchestrate development of functional beige fat. *Cell* **157**, 1292–1308 (2014).
2. Oguri, Y. *et al.* CD81 Controls Beige Fat Progenitor Cell Growth and Energy Balance via FAK Signaling. *Cell* **182**, 563–577 e520 (2020).

REVIEWER COMMENTS

Reviewer #1 (Remarks to the Author):

The authors have addressed my earlier comments partially.

However, I'm still far from being convinced about the conclusions obtained from the modeling. The information provided in the manuscript and the materials is insufficient to judge if the conclusions are valid or pure speculation. At the moment this part, and given that the authors did not try to address this carefully and did not provide the requested information, is pure speculation. Starting from the model generation, the authors need to provide a sequence alignment of PML and FOXO1/3 where they indicate which regions of FOXO1/3 they think are actually interacting. At the moment, not even an information about the amino acid regions of the constructs used in MD are provided. The quality of the SUMO-FOXO models generated as input for the MD calculations is essential as this part is highly prone to artefacts and could lead to invalid conclusions.

Reviewer #2 (Remarks to the Author):

The manuscript addressed a new and important finding regarding the function of FOXO1, all the questions are addressed by the authors and I have no more comments.

Reviewer #3 (Remarks to the Author):

This is a twice revised submission containing significant amount of work showing substantial improvement after the most recent submission. It is appreciated that the robustness of the work is more clearly demonstrated by the supplementation of data suggested by reviewers comments. Although improvements have been made and the results remain intriguing, there are still critical aspects of the study that could benefit from further attention. It is recognized and commended that the authors present a large number of models, however, lack of control groups causes difficulties in appreciating the robustness of each model individually. The explanation regarding the challenges in maintaining control groups for extended cold storage and the issues with islet auto-lysis and ESC detachment is well received. Including suitable controls would provide a better understanding of the extent of the presented results, even if this requires shortening the storage duration or adapting the conditions to prevent auto-lysis and detachment. A similar rationale can be followed regarding the control groups of the pre-diabetic mice cold exposure as well as human pancreatic islet experiments. Your intentions for future studies are much appreciated, however, to show that experimental conditions are in concordance

with literature would provide a more comprehensive understanding of the presented results.

Reviewer #4 (Remarks to the Author):

My concerns about this paper have been largely addressed and I am satisfied with the revisions made by the authors. In particular, I think it is an important finding that the inhibitor effects are primarily mediated by skeletal muscle rather than BAT.

However, I am not an expert on the concerns raised by Reviewer 1 regarding the molecular dynamics (MD) simulation, but I would only accept this manuscript if Reviewer 1 is satisfied with the authors' revisions regarding MD. Alternatively, I believe that the paper stands even if the MD simulation results (Figure 4F) are removed.

Response to reviewers' comments

In this 2nd revised version, we have highlighted major changes with cyan background. Some texts in 'Introduction' were missing and have been added and highlighted. Due to personnel changes in Ou's lab, two new authors were added as they carried out the experiments advised by Reviewer #3. Please see our point-to-point responses to your comments below.

REVIEWER COMMENTS

Reviewer #1 (Remarks to the Author):

The authors have addressed my earlier comments partially.

However, I'm still far from being convinced about the conclusions obtained from the modeling. The information provided in the manuscript and the materials is insufficient to judge if the conclusions are valid or pure speculation. At the moment this part, and given that the authors did not try to address this carefully and did not provide the requested information, is pure speculation. Starting from the model generation, the authors need to provide a sequence alignment of PML and FOXO1/3 where they indicate which regions of FOXO1/3 they think are actually interacting. At the moment, not even an information about the amino acid regions of the constructs used in MD are provided. The quality of the SUMO-FOXO models generated as input for the MD calculations is essential as this part is highly prone to artefacts and could lead to invalid conclusions.

Response:

As the lead contact of this manuscript, I'd apologize for omitting extra supporting information for the MD simulation. I was traveling while preparing documents for the long-overdue 1st revision, and it did not occur to me that extra information was missing. In this 2nd revision, Dr. Qingfen Yu has helped us to prepare a new 'Supplementary data 3' (Word document) and Supplementary file 'All_SIM_Seqs.fastq' to help explain how the MD simulation was carried out. Please also see the revised method description (see 'Supplementary Information', highlighted lines 210-215), together with Fig. 4f, Supplementary Fig. 6, and Table 5. We hope these documents and data have sufficiently addressed your concern and may earn your approval of our simulation modeling.

If from your expertise you are still not convinced regarding the simulation, as you may agree that our story has been well supported by lots of experimental data and been thoroughly reviewed by other experts, we would humbly ask your recommendation that allows us to withdraw the simulation data for future follow-up studies, and progress with the rest of the content like Reviewer #4 suggested.

Reviewer #2 (Remarks to the Author):

The manuscript addressed a new and important finding regarding the function of FOXO1, all the questions are addressed by the authors and I have no more comments.

Response:

We sincerely thank your approval!

Reviewer #3 (Remarks to the Author):

This is a twice revised submission containing significant amount of work showing substantial improvement after the most recent submission. It is appreciated that the robustness of the work is more clearly demonstrated by the supplementation of data suggested by reviewers comments. Although improvements have been made and the results remain intriguing, there are still critical aspects of the study that could benefit from further attention. It is recognized and commended that the authors present a large number of models, however, lack of control groups causes difficulties in appreciating the robustness of each model individually. The explanation regarding the challenges in maintaining control groups for extended cold storage and the issues with islet auto-lysis and ESC detachment is well received. Including suitable controls would provide a better understanding of the extent of the presented results, even if this requires shortening the storage duration or adapting the conditions to prevent auto-lysis and detachment. A similar rationale can be followed regarding the control groups of the pre-diabetic mice cold exposure as well as human pancreatic islet experiments. Your intentions for future studies are much appreciated, however, to show that experimental conditions are in concordance with literature would provide a more comprehensive understanding of the presented results.

Response:

After discussing with other authors of this work, and given the situation we are in (explain below), we have prepared additional data in this 2nd revision to address your remaining concerns:

- 1) Longer-time recovery of H1 ESC: in the 1st revision, you recommended us to investigate how H1 ESCs recover from cold exposure for a longer duration. Other than the detachment issue from extended cold exposure, we would like to bring to your attention that if the cold exposure is shorter, because H1 ESCs are relatively cold-resilient, there would not be obvious phenotypes to look for. Yet it seemed that you insisted we should do so. Hence, we figured out a suitable experiment to apply FOXO1 inhibiting drugs to weaken H1 ESC FOXO1 functions, and managed to observe phenotypes demonstrated in **Supplementary Fig. 1e**; please also see ‘manuscript’ highlighted lines 111-116, and ‘Supplementary Information’ highlighted lines 338-342.
- 2) ‘Control’ cold experiments with young healthy adult mice: as we understand you insisted that KPT-330 treatment and cold-exposure experiments with young healthy mice should be included as ‘control experiments’ in this work, new experiments were conducted, results summarized in **Supplementary Fig. 8g**, showing similar effects of KPT-330 in enhancing heat production, O₂ consumption and CO₂ production in young adult mice. Please also see ‘manuscript’ highlighted lines 248 and 317-330, and ‘Supplementary Information’ highlighted lines 466-470. Nonetheless, we’d like to

reiterate that we are very interested in the possible difference in cold-induced stress and thermogenic responses between young healthy and mid-age obese mice, and the roles of different organs (like Reviewer #4 suggested) in cold adaptation. We hope you would agree that these new results are sufficient for this story, and hence can allow us to initiate a new story in this direction.

- 3) Mouse islet OCR and GSIS experiments following 24-h *in vitro* recovery: we agreed with you in this regard. It seems your main concern is that our 2-h recovery time is too short, and data may not be directly comparable to that from others in the field. Following your suggestion and information from the references you previously listed, we tested islet OCR and GSIS following 5-day cold storage and 24-h *in vitro* recovery. Please see **Fig. 7c-f**, and ‘manuscript’ highlighted lines 269-276, and 735-740. As expected, mouse islets cold-stored in UW solution for 5 days had lost their ability to recover, whilst those stored in HS + KPT-330 could recover. Hopefully, these new results can make our data comparable to those from other published works and hence sufficiently address your concern.
- 4) Human pancreatic tissues and islets: we have to disclose to you and our editor that, as this project has lasted for quite a few years, we no longer had access to human pancreatic tissues as our protocol expired in 2023. Our hospital has implemented strict research rules on obtaining and usage of human donor organs and tissues. After reviewing our work and reviewers’ comments, our ethics committee think that this work shows adequate quality to be accepted by a good journal without the need to obtain more donor pancreases. Renewal of our donor pancreas protocol is only possible given the acceptance of this work and that the new project is more clinically oriented (i.e. more islet transplantation-focused trials). Hence, your understanding of our situation and help with the progression of this work will be much appreciated!

Reviewer #4 (Remarks to the Author):

My concerns about this paper have been largely addressed and I am satisfied with the revisions made by the authors. In particular, I think it is an important finding that the inhibitor effects are primarily mediated by skeletal muscle rather than BAT.

However, I am not an expert on the concerns raised by Reviewer 1 regarding the molecular dynamics (MD) simulation, but I would only accept this manuscript if Reviewer 1 is satisfied with the authors' revisions regarding MD. Alternatively, I believe that the paper stands even if the MD simulation results (Figure 4F) are removed.

Response:

I have explained to Reviewer #1 above why certain information was missing. Updated results and texts can be found in our response to Reviewer #1. ‘Manuscript’ highlighted lines 317-330 have been updated to reflect our self-awareness on the limits of the current study and our

interests in future pursuit. Also, your comment on an alternative solution if our MD simulation is not satisfactory to experts in that field is appreciated!

REVIEWERS' COMMENTS

Reviewer #1 (Remarks to the Author):

Thank you for providing the additional information. Given the poor sequence conservation, I'm still not convinced regarding the simulations. As suggested by the authors and Reviewer #4 I agree and recommend to withdraw the simulation data for future follow-up studies, and progress with the rest of the content.

Reviewer #3 (Remarks to the Author):

Thank you for addressing the comments from both reviews. The newly conducted experiments contribute to the robustness of the paper, especially the addition of control groups and longer recovery times prior to assessment. Regarding the human tissue, I appreciate your explanation. From a scientific point of view, I continue to believe that the data shown is limited to claim successful storage up to 14 days as only morphology images and insulin secretion are shown very early in the recovery process (at 2h).